# GLASS FLOWS: TRANSITION SAMPLING FOR ALIGNMENT OF FLOW AND DIFFUSION MODELS

**Peter Holderrieth**[1], **Uriel Singer**[2], **Tommi Jaakkola**[1], **Ricky T. Q. Chen**[2],
**Yaron Lipman**[2], **Brian Karrer**[2]
[1]MIT CSAIL, [2]FAIR, Meta

## ABSTRACT

The performance of flow matching and diffusion models can be greatly improved at inference time using reward alignment algorithms, yet efficiency remains a major limitation. While several algorithms were proposed, we demonstrate that a common bottleneck is the *sampling* method these algorithms rely on: many algorithms require to sample Markov transitions via SDE sampling, which is significantly less efficient and often less performant than ODE sampling. To remove this bottleneck, we introduce GLASS Flows, a new sampling paradigm that simulates a "flow matching model within a flow matching model" to sample Markov transitions. As we show in this work, this "inner" flow matching model can be retrieved from a pre-trained model without any re-training, combining the efficiency of ODEs with the stochastic evolution of SDEs. On large-scale text-to-image models, we show that GLASS Flows eliminate the trade-off between stochastic evolution and efficiency. Combined with Feynman-Kac Steering, GLASS Flows improve state-of-the-art performance in text-to-image generation, making it a simple, drop-in solution for inference-time scaling of flow and diffusion models.

## 1 INTRODUCTION

Flow matching and diffusion models have revolutionized the generation of images, videos, and many other data types (Lipman et al., 2022; Albergo et al., 2023; Liu et al., 2022; Song et al., 2020b; Ho et al., 2020). They convert Gaussian noise into realistic images or videos by simulating an ordinary or stochastic differential equation (ODE/SDE). Trained on large web-scale datasets, these models can generate highly realistic images or videos at unprecedented quality. Due to diminishing returns of pre-training these models, many recent works propose methods to improve models at inference-time, i.e. by optimizing additional objectives commonly referred to as *rewards* (Uehara et al., 2025). These **reward alignment** algorithms allow to enhance text-to-image alignment (Zhang et al., 2025), solve inverse problems (Chung et al., 2022; He et al., 2023), and improve molecular design (Li et al., 2025b). However, as these algorithms achieve higher performance at the expense of more compute, efficiency remains a major challenge for deploying reward alignment algorithms.

Inference of flow and diffusion models has so far followed one of two sampling paradigms: (1) ODE sampling, as used in flow matching or the "probability flow ODE" in diffusion models, and (2) SDE sampling. Empirically, it is well-known that ODE sampling is significantly more efficient and is therefore the main choice for deployment of large-scale models (Karras et al., 2022; Esser et al., 2024). However, a useful characteristic of SDE sampling is that it is *random*, i.e. a future point $X_{t'}$ is not determined by the present $X_t$ but characterized by transition probabilities

$$p_{t'|t}(x_{t'}|x_t) = \mathbb{P}[X_{t'} = x_{t'}|X_t = x_t], \quad (x_t, x_{t'} \in \mathbb{R}^d, 0 \le t < t' \le 1) \tag{1}$$

where $p_{t'|t}$ is called the **transition kernel**. Many reward alignment algorithms rely on sampling from $p_{t'|t}$. For example, search methods use samples $X_{t'} \sim p_{t'|t}(\cdot|x_t)$ as branches of a search tree (for ODE sampling, there would be only one branch). This creates a dilemma: So far, it is not known how to obtain samples $X_{t'} \sim p_{t'|t}(\cdot|x_t)$ using ODEs. Therefore, one has to switch from ODE to SDE sampling, losing efficiency in order to use an alignment algorithm that is meant to increase it.

In this work, we present a method to sample from transitions $p_{t'|t}$ using ODEs: **G**aussian **La**tent **Su**fficient **St**atistic (**GLASS**) Flows. GLASS Flows combine (1) the high efficiency of ODEs with (2) the controllable stochastic evolution characteristic of SDEs. GLASS Flows construct an "inner flow matching model" to sample from $p_{t'|t}$ (see fig. 1). Crucially, this inner flow matching can be

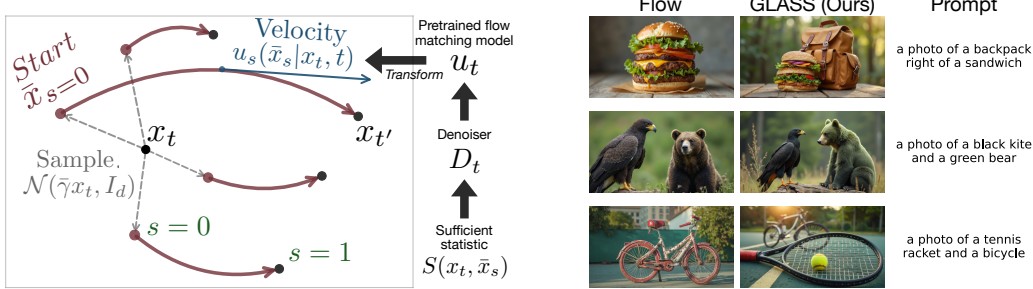

Figure 1: GLASS Flows overview. Left: Sampling transition $p_{t'|t}(x_{t'}|x_t)$ with GLASS Flows. Initial Gaussian samples $\bar{x}_{s=0}$ are evolved from inner time $s = 0$ to $s = 1$ via the velocity field $u_s(\bar{x}_s|x_t, t)$ that is obtained by transforming a pre-trained flow matching model. Right: Reward alignment with GLASS Flows improves text-image alignment.

easily obtained from pre-trained flow matching models **without any fine-tuning** - this transformation relies on the concept of a *sufficient statistic*, a fundamental tool in theoretical statistics (Fisher, 1922). Hence, GLASS Flows are a simple plug-in for any algorithm relying on SDE sampling. We apply GLASS Flows to reward alignment and improve the state-of-the-art performance in text-to-image generation. To summarize, we make the following contributions:

1. We introduce *GLASS Flows*, a method for efficiently sampling flexible Markov transitions via ODE's leveraging pre-trained flow and diffusion models.

2. We demonstrate that GLASS Flows can sample Markov transitions with significantly higher efficiency and lower discretization error than SDEs.

3. Text-to-image generation via GLASS Flows is shown to perform on par with ODE sampling, indicating GLASS Flows have eliminated the efficiency and stochasticity tradeoff.

4. We show significant performance improvements for text-to-image generation at zero cost by plugging GLASS Flows into Sequential Monte Carlo and reward guidance procedures.

## 2 BACKGROUND AND MOTIVATION

In this section, we introduce the necessary background on flow and diffusion models. We follow the flow matching (FM) framework (Lipman et al., 2022; Albergo et al., 2023; Liu et al., 2022), yet everything applies similarly to diffusion models (see appendix B.4). We denote data points with $z \in \mathbb{R}^d$ and the **data distribution** with $p_{\text{data}}$. Here, $t = 0$ corresponds to noise ($\mathcal{N}(0, I_d)$) and $t = 1$ to data ($p_{\text{data}}$). To noise data $z \in \mathbb{R}^d$, we use a **Gaussian conditional probability path** $p_t(x_t|z)$:

$$x_t = \alpha_t z + \sigma_t \epsilon, \quad \epsilon \sim \mathcal{N}(0, I_d) \quad \Leftrightarrow \quad p_t(x_t|z) = \mathcal{N}(x_t; \alpha_t z, \sigma_t^2 I_d) \tag{2}$$

where $\alpha_t, \sigma_t \geq 0$ are **schedulers** with $\alpha_0 = \sigma_1 = 0$ and $\alpha_1 = \sigma_0 = 1$ and $\alpha_t$ (resp. $\sigma_t$) strictly monotonically increasing (resp. decreasing) and continuously differentiable. With $z \sim p_{\text{data}}$ random, this induces a **marginal probability path** $p_t(x_t) = \mathbb{E}_{z \sim p_{\text{data}}}[p_t(x_t|z)]$ which interpolates Gaussian noise $p_0 = \mathcal{N}(0, I_d)$ and data $p_1 = p_{\text{data}}$. FM models learn the **marginal vector field**:

$$u_t(x_t) = \int u_t(x_t|z) p_{1|t}(z|x_t) dz, \quad p_{1|t}(z|x_t) = \frac{p_t(x_t|z) p_{\text{data}}(z)}{p_t(x_t)} \tag{3}$$

where $u_t(x_t|z)$ is the conditional vector field (see eq. (23) for formula). Simulating an ODE with the marginal vector field from initial Gaussian noise leads to a trajectory whose marginals are $p_t$:

$$X_0 \sim p_0, \quad \frac{d}{dt} X_t = u_t(X_t) \quad \Rightarrow \quad X_t \sim p_t \tag{4}$$

In particular, $X_1 \sim p_{\text{data}}$ returns a sample from the desired distribution. This sampling method is commonly called **ODE sampling** with a flow matching model. In the diffusion literature, sampling in this way is called the *probability flow ODE* (Song et al., 2021). In addition, one can also sample using the **time-reversal SDE** (Song et al., 2021) given by

$$X_0 \sim \mathcal{N}(0, I_d), \quad dX_t = \left[ u_t(X_t) + \frac{\nu_t^2}{2} \nabla \log p_t(X_t) \right] dt + \nu_t dW_t, \quad \nu_t^2 = 2 \frac{\dot{\alpha}_t}{\alpha_t} \sigma_t^2 - 2\sigma_t \dot{\sigma}_t \tag{5}$$

where $\nabla \log p_t(x_t)$ is the **score function** and $\dot{\sigma}_t = \partial_t \sigma_t, \dot{\alpha}_t = \partial_t \alpha_t$ are the time-derivatives of the schedulers (see eq. (33) for a derivation). As this is the limit process of DDPM (Ho et al., 2020), we refer to this as **DDPM sampling** in this work, regardless of the schedulers used. As the score function $\nabla \log p_t$ is just a reparameterization of $u_t$ (see appendix A.1), this SDE can be simulated using the same neural network. While every $\nu_t \geq 0$ results in a valid sampling procedure (Karras et al., 2022; Albergo et al., 2023; Lipman et al., 2024), we restrict ourselves to the choice of $\nu_t$ corresponding to the time-reversal SDE (DDPM sampling) as this is most commonly used.

## 3 MOTIVATION: EFFICIENT TRANSITIONS FOR REWARD ALIGNMENT

Inference-time reward alignment considers that the data distribution $p_{\text{data}}$ is not the "desired distribution" that the model should sample from. To align models better with our goals post-training, one uses $p_{\text{data}}$ only as a prior distribution and steers samples from the model to maximize a user-specified objective function $r : \mathbb{R}^d \to \mathbb{R}$ called the **reward function**. This goal is formalized as sampling from the **reward-tilted distribution**

$$p^r(z) = \frac{1}{Z^r} p_{\text{data}}(z) \exp(r(z)) \quad (Z_r > 0) \tag{6}$$

Note that the likelihood $p^r(z)$ is high if $p_{\text{data}}(z)$ is high *and* $r(z)$ is high. We briefly review three of the most common reward alignment algorithms and how they rely on stochastic transitions $p_{t'|t}$.

**Sequential Monte Carlo (SMC) methods** (Wu et al., 2023a; Singhal et al., 2025; Skreta et al., 2025) use a transition kernel $p_{t'|t}$ as a **proposal distribution**. They evolve $K$ particles $x_t^k$ via

$$x_{t'}^k \sim p_{t'|t}(\cdot|x_t^k) \quad (0 \leq t < t' \leq 1, k = 1, \ldots, K) \tag{7}$$

The particles are then evaluated via potentials $G(x_t, x_{t'})$ that guide the particles towards the desired tilted distribution, e.g. $G(x_t, x_{t'}) = \exp(r(x_{t'}) - r(x_t))$. Subsequently, the particles are resampled:

$$\underbrace{a_{t'}^k \sim \text{Multinomial}(G(x_t^1, x_{t'}^1), \cdots, G(x_t^K, x_{t'}^K))}_{\text{sample indices}}, \quad \underbrace{x_{t'}^k = x_{t'}^{a_{t'}^k}}_{\text{reassign particles}} \quad (k = 1, \cdots, K)$$

Here, SMC sequentially replaces "unpromising" particles by "promising" ones.

**Search methods** (Li et al., 2025b; Zhang et al., 2025) consider DDPM sampling as a rollout of a search tree with branches coming from samples from $p_{t'|t}$. Beyond sampling branches of the search tree, search methods use approximations of the **value function** (Li et al., 2025b) defined via

$$V_t(x_t) = \log \mathbb{E}_{z \sim p_{1|t}(\cdot|x_t)}[\exp(r(z))], \quad \text{where } p_{1|t}(z|x_t) = p_t(x_t|z)p_{\text{data}}(z)/p_t(x_t) \tag{8}$$

to evaluate nodes, i.e. to select nodes in the tree. Estimating the value function $V_t$ relies on the **flow matching posterior** $p_{1|t}$. This is a special case of a DDPM transition (Song et al., 2020b):

$$p_{1|t}(z|x_t) = p_{t'=1|t}^{\text{DDPM}}(X_1 = z|X_t = x_t) \tag{9}$$

As sampling from $p_{1|t}(z|x_t)$ is only possible with the SDE so far and therefore inefficient, most search methods use approximations of this function (Li et al., 2025b; Zhang et al., 2025). Similarly, approximations of the value function $V_t(x_t)$ can also be used to define potentials in SMC procedures.

**Guidance methods.** Guidance methods (Skreta et al., 2025; Chung et al., 2022; He et al., 2023; Feng et al., 2025) modify the vector field $u_t$ of the flow matching or diffusion model using an intermediate reward function $r_t : \mathbb{R}^d \to \mathbb{R}$ such that $r_1(z) = r(z)$:

$$u_t^r(x) = u_t(x) + c_t \nabla r_t(x) \quad (c_t \geq 0) \tag{10}$$

Again, ideally $r_t(x) = V_t(x)$, which is computationally heavy to estimate for the same reasons. Instead, one can define $r_t(x)$ via simple approximations and potentially correct using SMC and SDE sampling (see e.g. (Skreta et al., 2025, Proposition 3.4)).

**GLASS Flows motivation.** Instead of proposing another reward alignment algorithm, we take a complementary approach: We optimize the transitions these methods rely on. Specifically, we aim to (1) improve how to sample the transitions; and (2) extend the space of transitions that we can sample from. As most deployed models use ODE sampling for efficiency, the reliance of inference-time reward alignment on stochastic transitions from SDEs is a common handicap making them slower and less performant. This motivates our goals:

**Goal 1:** Simulate transitions $p_{t'|t}$ in an *efficient* way (via ODEs); and such that they are *stochastic* (i.e. emulates DDPM sampling).

**Goal 2:** Extend the space of transitions $p_{t'|t}$ to allow for more effective reward alignment (e.g. SMC or search).

## 4 GLASS FLOWS

In this section, we present *GLASS Flows*, a novel way of sampling transitions from pre-trained flow and diffusion models. We begin by explaining the core idea.

Let us be given a point $X_t = x_t$ in a flow matching or diffusion trajectory. Given a time $t' > t$, our goal is to sample $X_{t'} \sim p_{t'|t}(x_{t'}|x_t)$ from a transition kernel $p_{t'|t}$. We can consider this as a conditional generative modeling problem in itself. In other words, the variables $x_t, t$ are the variables we condition on (i.e. "prompts") and we want to sample $x_{t'}$. To do this, we can, in turn, construct an **inner flow matching model** $u_s(\bar{x}_s|x_t, t)$ with a **new time variable** $s$ ($0 \le s, t \le 1, \bar{x}_s, x_t \in \mathbb{R}^d$) that is supposed to model the transition kernel of $p_{t'|t}$. Specifically, we want to construct $u_s(\bar{x}_s|x_t, t)$ such that after sampling from this model via

$$\bar{X}_0 \sim p_{\text{init}}, \quad \frac{\mathrm{d}}{\mathrm{d}s}\bar{X}_s = u_s(\bar{X}_s|x_t, t) \quad \Rightarrow \bar{X}_1 \sim p_{t'|t}(\cdot|X_t = x_t) \tag{11}$$

we get samples from the transition kernel $p_{t'|t}$ at $s = 1$ for an appropriate initial distribution $p_{\text{init}}$. Note that with this approach, we achieve stochasticity by sampling the inner initial condition $\bar{X}_0$, while the subsequent evolution follows a deterministic ODE. In contrast, SDE transitions have deterministic initial conditions but the increments are stochastic. We present a simple algorithm to obtain $u_s(\bar{x}_s|x_t, t)$ that we explain in this section (see algorithm 1).

---

**Algorithm 1** Transition sampling with GLASS Flows (with Euler ODE integration)

1: **def** $D(x_t, t)$: ▷ FM denoiser
2:     $u \leftarrow u_t(x_t)$ ▷ neural net call
3:     **return** $\frac{1}{\dot{\bar{\alpha}}_t \sigma_t - \alpha_t \dot{\sigma}_t}(\sigma_t u - \dot{\sigma}_t x_t)$
4:
5: **def** $D(x_t, \bar{x}_s, \mu, \Sigma)$: ▷ GLASS denoiser
6:     **If** $s = 0$: **return** $D_t(x_t)$
7:     $S(\mathbf{x}) \leftarrow \frac{\mu^T \Sigma^{-1}}{\mu^T \Sigma^{-1}\mu}[x_t, \bar{x}_s]^T$
8:     $t^* \leftarrow g^{-1}((\mu^T \Sigma^{-1}\mu)^{-1})$
9:     **return** $D(\alpha_{t^\star} S(\mathbf{x}), t^\star)$
10:
11: **def** $u_s(\bar{x}_s|x_t, t)$: ▷ GLASS velocity
12:     $\Sigma \leftarrow \begin{bmatrix} \sigma_t^2 & \sigma_t^2\bar{\gamma} \\ \sigma_t^2\bar{\gamma} & \bar{\sigma}_s^2 + \bar{\gamma}^2\sigma_t^2 \end{bmatrix}, \mu \leftarrow \begin{bmatrix} \alpha_t \\ \bar{\alpha}_s + \bar{\gamma}\alpha_t \end{bmatrix}$
13:     $\hat{z} \leftarrow D(x_t, \bar{x}_s, \mu, \Sigma)$
14:     $w_1 \leftarrow \frac{\partial_s \bar{\sigma}_s}{\bar{\sigma}_s}; w_2 \leftarrow \partial_s\bar{\alpha}_s - \bar{\alpha}_s w_1; w_3 \leftarrow -\bar{\gamma}w_1$
15:     **return** $w_1\bar{x}_s + w_2\hat{z} + w_3 x_t$
16:

**Input:** Start time $t$, end time $t'$, current position $x_t$, pre-trained FM model $u_t$, schedulers $\alpha_t, \sigma_t, \bar{\alpha}_s, \bar{\sigma}_s$, correlation $\rho$, number of steps $M$

**Output:** Sample $X_{t'} \sim p_{t'|t}(x_{t'}|x_t)$

1: $\bar{\gamma} \leftarrow \rho\,\sigma_{t'}/\sigma_t$
2: Sample $\epsilon \sim \mathcal{N}(0, I_d)$
3: $\bar{X}_0 \leftarrow \bar{\gamma}\,x_t + \bar{\sigma}_0\epsilon$
4: $s \leftarrow 0$
5: $h \leftarrow 1/M$
6: **for** $m = 0, \ldots, M-1$ **do**
7:     $v \leftarrow u_s(\bar{X}_s|x_t, t)$ ▷ Call function
8:     $\bar{X}_s \leftarrow \bar{X}_s + hv$
9:     $s \leftarrow s + h$
10: **end for**
11: **Return** $\bar{X}_1$

---

### 4.1 GLASS TRANSITIONS

We first define the family of transitions $p_{t'|t}$ to sample from. To define a transition kernel $p_{t'|t}$ in a flow matching model, we want $X_t, X_{t'}$ to have marginals given by the probability path:

$$X_t \sim \mathcal{N}(\alpha_t z, \sigma_t^2 I_d), \quad X_{t'} \sim \mathcal{N}(\alpha_{t'} z, \sigma_{t'}^2 I_d) \quad (z \sim p_{\text{data}})$$

Therefore, given a data point $z \in \mathbb{R}^d$, the respective mean and variances of $X_t, X_{t'}$ are fixed. However, we have a degree of freedom to set the correlation $\rho$ between $X_t$ and $X_{t'}$. Specifically, we define mean scale $\mu$ and covariance $\Sigma$ as

$$\mu = \begin{pmatrix} \mu_1 \\ \mu_2 \end{pmatrix} = \begin{pmatrix} \alpha_t \\ \alpha_{t'} \end{pmatrix}, \quad \Sigma = \begin{pmatrix} \Sigma_{11} & \Sigma_{12} \\ \Sigma_{21} & \Sigma_{22} \end{pmatrix} = \begin{pmatrix} \sigma_t^2 & \rho\sigma_t\sigma_{t'} \\ \rho\sigma_t\sigma_{t'} & \sigma_{t'}^2 \end{pmatrix}$$

where the **correlation** $-1 \leq \rho \leq 1$ is the degree of freedom that we can choose. Then let us define the tuple $\mathbf{X} = (X_t, X_{t'})^T$ and define the joint distribution as

$$\mathbf{X} \sim p_{t,t'}(\mathbf{X}|z) = \prod_{j=1}^{d} \mathcal{N}\left((X_t^j, X_{t'}^j); z^j \mu, \Sigma\right) \quad (z = (z^1, \cdots, z^d)^T \sim p_{\text{data}}) \tag{12}$$

Each coordinate is noised identically and independently - we only allow for correlations across time, not across coordinates. Every joint distribution $p_{t',t}(X_t, X_{t'})$ also defines a conditional distribution

$$p_{t'|t}(X_{t'}|X_t) = \frac{p_{t,t'}(X_t, X_{t'})}{p_t(X_t)} \qquad \blacktriangleright \text{ GLASS transition} \tag{13}$$

which defines the **GLASS transition**. This is a large family of transitions where $\rho$ controls the similarity between $X_t$ and $X_{t'}$. It includes the important example of DDPM transitions:

**Proposition 1.** For $\rho = \frac{\alpha_t \sigma_{t'}}{\sigma_t \alpha_{t'}}$, we get that: $p_{t'|t}^{\text{DDPM}}(X_{t'}|X_t) = p_{t'|t}(X_{t'}|X_t)$, i.e. DDPM transitions are a special case of GLASS transitions.

See appendix A.2 for a proof. Note that $\rho$ defined like this is a valid correlation coefficient (i.e. $|\rho| \leq 1$) because $\frac{\sigma_{t'}}{\sigma_t} < 1$ and $\frac{\alpha_t}{\alpha_{t'}} < 1$ by monotonicity of the schedulers.

## 4.2 Constructing the velocity field

In this section, we show how to construct $u_s(\bar{x}_s|x_t, t)$ to sample from the GLASS transition $p_{t'|t}$ from pre-trained flow matching and diffusion models **without any re-training or fine-tuning**. A fundamental concept we use is a *denoiser model* $D_t$ defined as the expectation of the posterior:

$$D_t(x) = \int z p_{1|t}(z|x) \mathrm{d}z = \frac{1}{\dot{\alpha}_t \sigma_t - \alpha_t \dot{\sigma}_t}(\sigma_t u_t(x_t) - \dot{\sigma}_t x_t). \tag{14}$$

The second equation shows that we can easily obtain the denoiser by reparameterizing the velocity field $u_t$ (see appendix A.1 for derivation). In the following, we use the same reparameterization idea but the other way around: To construct $\bar{u}_s(\bar{x}_s|x_t, t)$, we (1) derive a denoiser model for Markov transitions and (2) reparameterize it to obtain the velocity field $\bar{u}_s(\bar{x}_s|x_t, t)$.

### 4.2.1 GLASS denoiser

We begin by extending the idea of a denoiser to Markov transitions from $x_t$ to $x_{t'}$. In eq. (12), we have defined a joint distribution over $\mathbf{X} = (X_t, X_{t'})$ specified by some mean scale $\mu$ and covariance $\Sigma$. Therefore, we define the **GLASS denoiser** as the expected posterior given both $x_t$ *and* $x_{t'}$:

$$D_{\mu,\Sigma}(\mathbf{x}) = \int z p(Z = z|\mathbf{X} = \mathbf{x}) \mathrm{d}z, \quad \mathbf{x} = (x_t, x_{t'}), x_t, x_{t'} \in \mathbb{R}^d \tag{15}$$

Here, it is instructive to think of $x_t$ as a noisy measurement of a parameter $z$. The "standard" denoiser $D_t$ represents the mean of $z$ given *one* Gaussian measurement $x_t$, while the GLASS denoiser $D_{\mu,\Sigma}$ represents the mean of $z$ given *two* Gaussian measurements $(x_t, x_{t'})$. Our core idea is that we can effectively "summarize" two measurements $(x_t, x_{t'})$ into a single variable via the transformation

$$S(\mathbf{x}) = \frac{\mu^T \Sigma^{-1} \mathbf{x}}{\mu^T \Sigma^{-1} \mu}, \quad \mathbf{x} = (x_t, x_{t'})^T \in \mathbb{R}^{2 \times d} \qquad \blacktriangleright \text{ sufficient statistic}$$

In theoretical statistics, $S(\mathbf{x})$ is called a **sufficient statistic** (Fisher, 1922; Casella & Berger, 2024), describing the idea that $S(\mathbf{x})$ carries as much information about the latent $Z = z$ as does $\mathbf{x}$. This is intuitive: $S(\mathbf{x})$ is a weighted average of $x_t, x_{t'}$ - the weight is higher the more informative an element is about $z$ (lower variance and higher scale factor $\mu$). Finally, define invertible function $g(t) = \sigma_t^2/\alpha_t^2$ as the effective noise scale defined by flow matching schedulers $\alpha_t, \sigma_t$. We get:

**Proposition 2.** Let $\mathbf{x} = (x_1, x_2)$ with $\mathbf{x}_i \in \mathbb{R}^d$ and $t^* = t^*(\mu, \Sigma) = g^{-1}((\mu \Sigma^{-1} \mu)^{-1})$. Then:

$$\underbrace{D_{\mu,\Sigma}(\mathbf{x})}_{\text{GLASS denoiser}} = \underbrace{D_{t^*}(\alpha_{t^*} S(\mathbf{x}))}_{\text{"standard" pre-trained denoiser with reparameterized input and time}}$$

where $D_t$ is defined as in eq. (14) and $\alpha_t$ is the scheduler in eq. (2).

So, the GLASS denoiser can be obtained by a single function evaluation of a pre-trained model (see algorithm 1). See appendix A.3 for a proof. We note that $g^{-1}$ is a simple analytical formula depending on the choice of $\alpha_t, \sigma_t$ (see appendix B.6 for specific formulas).

#### 4.2.2 GLASS VELOCITY FIELD

We now derive the GLASS velocity field $u_s(\bar{x}_s|x_t, t)$ as a reparameterization of the GLASS denoiser. Since $p_{t,t'}(x_t, x_{t'}|z)$ is Gaussian (see eq. (12)), also the conditional distribution is Gaussian

$$p_{t'|t}(x_{t'}|x_t, z) = \mathcal{N}(x_{t'}; \bar{\alpha}z + \bar{\gamma}x_t, \bar{\sigma}^2 I_d) \tag{16}$$

$$\text{where} \quad \bar{\gamma} = \rho\sigma_{t'}\sigma_t^{-1}, \quad \bar{\alpha} = \alpha_{t'} - \bar{\gamma}\alpha_t, \quad \bar{\sigma}^2 = \sigma_{t'}^2(1 - \rho^2) \tag{17}$$

Therefore, we can construct a conditional and marginal Gaussian probability path by

$$p_s(\bar{x}_s|x_t, z) = \mathcal{N}(\bar{x}_s; \bar{\alpha}_s z + \bar{\gamma}x_t, \bar{\sigma}_s^2 I_d), \quad p_s(\bar{x}_s|x_t) = \int p_s(\bar{x}_s|x_t, z)p_{1|t}(z|x_t)\mathrm{d}z \tag{18}$$

for schedulers $\bar{\alpha}_s, \bar{\sigma}_s$ such that $\bar{\alpha}_0 = 0, \bar{\alpha}_1 = \bar{\alpha}, \bar{\sigma}_1 = \bar{\sigma}, \bar{\sigma}_0^2 > 0$. These conditions ensure that the marginal probability path interpolates noise and the GLASS transition:

$$s = 0: \quad p_0(\bar{x}_0|x_t) = \mathcal{N}(\bar{x}_0; \bar{\gamma}x_t, \bar{\sigma}_0^2 I_d), \quad s = 1: \quad p_1(\bar{x}_1|x_t) = p_{t'|t}(X_{t'} = \bar{x}_1|x_t)$$

A natural choice of schedulers are ones such that $p_s(\bar{x}_s|x_t, z)$ is the optimal transport path (**CondOT schedulers** (Lipman et al., 2022)), i.e. $\bar{\alpha}_s = s\bar{\alpha}, \bar{\sigma}_s = (1 - s)\bar{\sigma}_0 + s\bar{\sigma}$. We present the following result (see appendix A.4 for proof):

> **Theorem 1.** Let us be given two times $t < t'$, a starting point $x_t$, and a correlation parameter $\rho$ defining the GLASS transition $p_{t'|t}$ in eq. (13). Then we can sample from $p_{t'|t}(\cdot|x_t)$ as follows:
>
> Define the **GLASS velocity field** as the weighted sum of $\bar{x}_s, x_t$ and the GLASS denoiser
>
> $$u_s(\bar{x}_s|x_t, t) = w_1(s)\bar{x}_s + w_2(s)D_{\mu(s),\Sigma(s)}(x_t, \bar{x}_s) + w_3(s)x_t \tag{19}$$
>
> with weight coefficients $w_1(s), w_2(s), w_3(s) \in \mathbb{R}$ and time-dependent mean scale and covariance $\mu(s), \Sigma(s)$ given by
>
> $$\mu(s) = \begin{pmatrix} \alpha_t \\ \bar{\alpha}_s + \bar{\gamma}\alpha_t \end{pmatrix}, \quad \Sigma(s) = \begin{pmatrix} \sigma_t^2 & \sigma_t^2\bar{\gamma} \\ \sigma_t^2\bar{\gamma} & \bar{\sigma}_s^2 + \bar{\gamma}^2\sigma_t^2 \end{pmatrix} \tag{20}$$
>
> $$w_1(s) = \frac{\partial_s\bar{\sigma}_s}{\bar{\sigma}_s}, \quad w_2(s) = \partial_s\bar{\alpha}_s - \bar{\alpha}_s w_1(s), \quad w_3(s) = -\bar{\gamma}w_1(s) \tag{21}$$
>
> where $\bar{\alpha}_s, \bar{\sigma}_s, \bar{\gamma}$ are chosen as in eq. (18). Then the final point $\bar{X}_1$ of the trajectory $\bar{X}_s$ obtained via the ODE
>
> $$\bar{X}_0 \sim \mathcal{N}(\bar{\gamma}x_t, \bar{\sigma}_0^2 I_d), \quad \frac{\mathrm{d}}{\mathrm{d}s}\bar{X}_s = u_s(\bar{X}_s|x_t, t) \tag{22}$$
>
> is a sample from the GLASS transition, i.e. $\bar{X}_1 \sim p_{t'|t}(\cdot|x_t)$. More generally, $\bar{X}_s \sim p_s(\cdot|x_t)$ for all $0 \leq s \leq 1$.

This theorem shows that any flow matching or diffusion model contains an "inner" flow matching model $u_s(\bar{x}_s|x_t, t)$ that allows to sample GLASS transitions. By proposition 2, no further training is required. As this result relies on the idea of using the sufficient statistic of Gaussian measurements to infer a latent $z$, we coin these flows **G**aussian **La**tent **S**ufficient **S**tatistic (**GLASS**) **Flows**. In algorithm 1, we describe pseudocode to sample a transition with GLASS Flows. Note that the reparameterizations and $2 \times 2$ matrix inversions are negligible compared to neural network evaluations. Therefore, the complexity of algorithm 1 is governed by the number of function evaluations of the pre-trained velocity field $u_t(x)$, i.e. the number of simulation steps $M$.

**Sampling with GLASS Flows.** To generate a data point $X_1 \sim p_{\text{data}}$, we set the number $K$ of transitions and transition times $0 \leq t_0 < t_1 < \cdots < t_K = 1$ and initialize $X_0 = X_{t_0} \sim \mathcal{N}(0, I_d)$. For every $k = 0, \cdots, K - 1$, we sample the transition from $X_{t_k}$ to $X_{t_{k+1}}$ using algorithm 1 (set $t = t_k$ and $t' = t_{k+1}$) with a choice of a correlation parameter $\rho$ that we can choose freely (note that it can also vary across transitions). Assuming no discretization error and perfect training, if $X_{t_k}$ is distributed according to the probability path, also the next step will have marginals specified by the probability path by theorem 1:

$$X_{t_k} \sim p_{t_k} \quad \Rightarrow \quad X_{t_{k+1}} \sim p_{t_{k+1}}$$

In particular, it will hold that $X_{t_K} = X_1 \sim p_1 = p_{\text{data}}$, i.e. the endpoint is a valid sample from the desired distribution. This preservation of marginals holds for any $\rho$ (not limited to the one corresponding to DDPM transitions). Therefore, GLASS is a novel sampling scheme resulting in Markov chains preserving the marginals of a pre-trained flow or diffusion models. The total number of function evaluations is $K \cdot M$. For $K = 1$, only one transition, we recover standard flow matching as the conditioning for $t_0 = 0$ is simply ignored. Further, for $M = 1$, one simulation step for the inner transition, we recover **DDIM sampling** (Song et al., 2020a) (see appendix B.2 for a derivation).

**Implementation.** For numerical stability, we need to account for the cases when $s = 0$ in algorithm 1. We derive this edge case in appendix B.3. In appendix B.3, we also discuss other techniques to make the implementation numerically stable. Further, all current large-scale flow matching models use **classifier-free guidance (CFG)** (Ho & Salimans, 2022) to condition on a prompt $c$. To use CFG with GLASS Flows, we treat the classifier-free guidance vector field $u_t^w(x|c) = (1 + w)u_t(x|c) - wu_t(x)$ as the ground truth vector field for the same weight $w \geq 0$, i.e. all calculations are done with this vector field. Finally, it is well-known that there are many **equivalent parameterizations of the vector field** $u_t$ (e.g. via the score function or directly via the denoiser) and also **diffusion models in discrete time**. We discuss in appendix B.4 how to construct GLASS Flows with these alternative parameterizations. Further, we provide a minimal implementation of algorithm 1 at `github.com/PeterHolderrieth/glass_flows_tutorial`.

### 4.3 INFERENCE-TIME REWARD ALIGNMENT WITH GLASS

Finally, we briefly explain how GLASS Flows can be applied to inference-time reward alignment, focusing on the algorithms discussed in section 3:

**Sequential Monte Carlo:** we use GLASS Flows to evolve the particles with the proposal distribution $p_{t'|t}$ (see eq. (7)) replacing SDE sampling (Singhal et al., 2025) with GLASS Flows.

**Value function estimation:** we estimate the value function $V_t(x_t)$, as used in search methods (see proposition 1), via samples from the posterior $p_{1|t}$ replacing SDE sampling with GLASS Flows.

**Reward guidance:** we can adjust the GLASS velocity field, analogous to eq. (10), to apply GLASS Flows with reward guidance. Specifically, we add an appropriately scaled gradient of an analogous value function derived in appendix B.1.

## 5 RELATED WORK

We discuss the most closely related work in this section and refer to appendix C for an extended discussion of other related methods. GLASS Flows operate in discrete-time, leveraging an underlying continuous-time model. Discrete-time diffusion (Sohl-Dickstein et al., 2015) appeared prior to continuous-time diffusion, but such models are parameterized as 1st order approximations of the same ODE/SDE and are not qualitatively different. GLASS Flows instead consider sampling from latent Gaussian transitions for arbitrarily distant times. Recently, discrete-time transitions in FM models were also studied in Transition Matching (Shaul et al., 2025). In fact, the DTM supervision process in Transition Matching (see (Shaul et al., 2025, equ. (10))) corresponds to a GLASS transition with $\rho = 1$ (see appendix C for detailed discussion). However, note that Transition Matching (Shaul et al., 2025) modifies pre-training and network architectures via patch approximations to sample transitions via flows, while our method focuses on inference-time modification post-training. Therefore, GLASS Flows and TM address different problems and lead to different models that are theoretically related, yet practically different.

Inference-time reward alignment methods are reviewed in (Uehara et al., 2025). They can be categorized into single particle (i.e. guidance) and more general multi-particle methods (i.e. Sequential Monte Carlo (SMC) and search). Guidance such as (Chung et al., 2022; Song et al., 2023b; Ye et al., 2024; Yu et al., 2023; Bansal et al., 2023; He et al., 2023; Song et al., 2023a; Feng et al., 2025) aims to approximate the difference between an existing velocity and an optimal velocity trained with a reward. In addition to guidance, source-based methods specific to flows keep the velocity fixed while altering the input noise distribution (Ben-Hamu et al., 2024; Eyring et al., 2024; Wang et al., 2025). Multi-particle methods evolve a population of particles such as SMC (Singhal et al., 2025; Skreta et al., 2025; Wu et al., 2023a; Mark et al., 2025; He et al., 2025) and search (Li et al., 2025b; Zhang et al., 2025).

Recently, Chen et al. (2025b) propose Training-free Augmented Dynamics (TADA) introducing training-free improvements of diffusion models using a similar mathematical principle as in this work. Specifically, they show that several recently proposed diffusion models with augmented state spaces (Dockhorn et al., 2021; Chen et al., 2023) can be recovered from a pre-trained FM or diffusion model using a Gaussian conjugacy/sufficient statistic argument (Chen et al., 2025b, Proposition 3.1). Further, this principle can be extended to state spaces augmented with more than 2 variables. This allows them to accelerate sampling significantly. While we design a different method designed to get fast stochastic transition samplers for reward alignment, both (Chen et al., 2025b) and this work use the same mathematical principles to derive their respective algorithms. We discuss this in more detail in appendix C.

Finally, reward fine-tuning methods based on GRPO (Xue et al., 2025; Li et al., 2025a; Liu et al.), stochastic optimal control (Liu et al.; Domingo-Enrich et al., 2024), DPO (Wallace et al., 2024) or other reinforcement learning approaches aim to achieve the same goal as this work, i.e. to align a diffusion model with a reward function $r$. GLASS has 2 different important synergies with these models: (1) Many of these algorithms require the use of DDPM/SDE sampling for exploration during training (Liu et al., 2025; Xue et al., 2025; Li et al., 2025a; Domingo-Enrich et al., 2024). However, this is very inefficient - as discussed in this work. Therefore, one could potentially accelerate reward fine-tuning methods via GLASS Flows by replacing the slow SDE sampling with GLASS Flows. (2) One can also apply inference-time scaling with GLASS Flows to fine-tuned models. Many of these models learn a FM model of reward-tilted distribution. Hence, GLASS Flows is equally valid to be applied to reward fine-tuned models and can be used to improve these models as well. Therefore, both approaches complement each other.

## 6 EXPERIMENTS

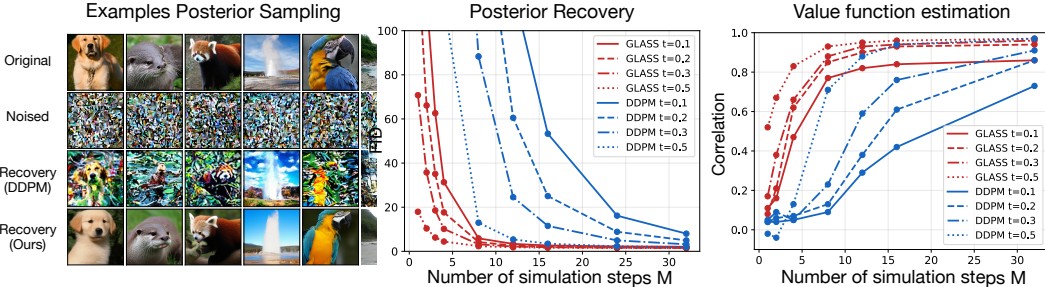

Figure 2: Posterior sampling experiments. We noise images and then sample from the posterior $z \sim p_{1|t}(\cdot|x)$ via DDPM or GLASS Flows. Left: Examples for $t = 0.2$ and $M = 6$ simulation steps. Middle: FID values for various simulation steps $M$ and time $t$. Right: Estimation of the value function as assessed by correlation with ground truth (200 Monte Carlo samples with $M = 200$).

### 6.1 EFFICIENT POSTERIOR SAMPLING AND VALUE FUNCTION ESTIMATION

**Posterior sampling.** We begin by benchmarking the efficiency of sampling transitions with GLASS Flows (our method) vs SDEs (DDPM sampling). As an example transition of particular importance, we use the posterior $p_{1|t}$ of the probability path (see eq. (9)). We use DiT/SiT models from (Peebles & Xie, 2023; Ma et al., 2024), a competitive class-conditional flow matching model, trained on ImageNet256. Our experimental setup is as follows: We sample data points from the ImageNet model ($z \sim p_{data}$), noise them ($x \sim p_t(\cdot|z)$), and then sample from the posterior via each respective method ($z' \sim p_{1|t}(\cdot|x)$). For many simulation steps ($M = 200$ in algorithm 1), both GLASS Flows and DDPM sampling give high quality samples from the posterior (see app. fig. 10). We then vary the number of simulation steps $M$ to values ranging from $M = 2$ to $M = 50$ and the time $t$. Note that the lower $M$ and the lower the time $t$, the "harder" the task gets as we have more discretization error and have added more noise to the reference image. Figures 11 to 13 show that GLASS Flows return significantly higher quality samples for low $M$ or $t$. To quantify this, we measure the image quality via Frechet Inception Distance (FID) using 50k images for both reference and each method (for each combination of $t$ and $M$). GLASS Flows achieve significantly better FID than DDPM sampling for the same number of sampling steps (see fig. 2). Therefore, **GLASS Flows represent a significant boost in efficiency when sampling from the posterior $p_{1|t}$.**

**Value function estimation.** Next, we investigate whether better sampling from $p_{1|t}$ also translates to better estimation of the value function $V_t$ (see eq. (8)). As a reward model, we use log-likelihoods of a ResNet ImageNet classifier (He et al., 2016). We repeat the same experiment, i.e. noise image and sample from the posterior, but this time measure the correlation (or MSE) between the ground truth value function and estimators. The ground truth is measured by using $M = 200$ simulation steps with the ODE/SDE and 200 samples. As one can see in fig. 2, GLASS Flows achieve significantly higher correlation for lower number of simulation steps. This demonstrates that the improved posterior via GLASS Flows translates to significantly better estimation of the value function.

## 6.2 NOVEL SAMPLING METHODS

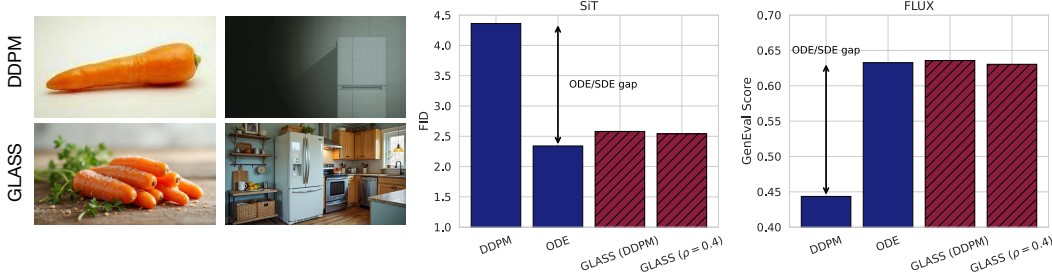

Figure 3: Sampling from SiT/FLUX with various sampling methods. Left: Comparison with FLUX of images generated with DDPM vs. GLASS Flows. DDPM samples are more blurry and of lower quality. Middle: Results for SiT. Right: Results for FLUX. Prompts: "Carrots" and "Refrigerator".

We next investigate how GLASS Flows perform as a novel scheme to sample from flow matching and diffusion models. We use the DiT/SiT models and the FLUX model (Labs, 2024), a state-of-the-art text-to-image model generating high resolution images (size $768 \times 1360$). We use 50 neural network evaluations (default for FLUX model) for all methods. For GLASS Flows, we use equally spaced $N = 6$ transition points. As one can see in fig. 3, ODE sampling vs. DDPM sampling have a significant performance gap for the default FLUX parameters. However, GLASS Flows close this gap both for SiT on ImageNet (FID) and the GenEval benchmark on FLUX. In fact, GLASS Flows perform on par with ODE sampling, while having stochastic transitions. Therefore, these results demonstrate that **GLASS Flows effectively remove the trade-off between efficiency and stochasticity for sampling**.

In appendix D.3, we perform an ablation experiment over various correlation parameters $\rho$. Overall, we find that all values of $\rho$ are numerically stable and lead to ODE-level performance with only minor differences. The choice of a constant correlation schedule of $\rho = 0.4$ led to the best results for the FLUX model and we use this in subsequent experiments. We note that the optimal choice of $\rho$ is dependent on the data and model and may well differ for other settings.

## 6.3 SEQUENTIAL MONTE CARLO EXPERIMENTS

Next, we apply GLASS Flows to inference-time reward alignment via Sequential Monte Carlo (SMC), in particular Feynman-Kac Steering (FKS) (Singhal et al., 2025; Skreta et al., 2025). We apply FKS on text-to-image generation using the FLUX model. We use a different pre-trained model than (Singhal et al., 2025) because FLUX is the current state-of-the-art model and our method requires a continuous-time model (the $t^*$ map does not fall into a discrete set of grid points). Except that, we use the same hyperparameters as in (Singhal et al., 2025). Between resampling steps, we sample the transitions with either DDPM sampling like previous works (Singhal et al., 2025) or GLASS Flows. We also compare against a Best-of-N baseline (BoN), i.e. where $N$ images are sampled and the one with highest reward selected. To ensure that we do not overfit to a single reward model, we run experiments for 4 different reward models (CLIP (Hessel et al., 2021), Pick (Kirstain et al., 2023), HPSv2 (Wu et al., 2023b), ImageReward (Xu et al., 2023a)). Further, to avoid "reward hacking", we evaluate results also on GenEval (Ghosh et al., 2023), to measure whether we can effectively optimize a reward without sacrificing GenEval performance. In table 1, we summarize our results and we plot examples in fig. 14. The first observation we make is that FKS with vanilla SDE sampling does not outperform a simple Best-of-N baseline as the Best-of-N is sampled with

Table 1: Sequential Monte Carlo via Feynman-Kac steering (FKS). Every reward model defines a new experiment whose samples we evaluate on the same reward model and the GenEval benchmark. We set $N = 8$ (number of particles). NFEs=400 for all rows except flow baseline (50 NFEs). BoN: Best-of-N. FKS: Feynman-Kac Steering.

| Algorithm | CLIP | | Pick | | HPSv2 | | IR | |
|---|---|---|---|---|---|---|---|---|
| | CLIP | GenEv. | Pick | GenEv. | HPSv2 | GenEv. | IR | GenEv. |
| Flow baseline | 34.9 | 63.2 | 23.4 | 63.2 | 0.302 | 63.2 | 0.88 | 63.2 |
| BoN-SDE | 36.9 | 60.8 | 23.1 | 63.5 | 0.303 | 60.9 | 1.16 | 65.3 |
| BoN-ODE | 38.5 | 70.6 | 23.8 | 69.3 | 0.315 | 69.8 | 1.31 | 71.8 |
| FKS-SDE (DDPM) | 39.0 | 64.1 | 23.4 | 63.0 | 0.295 | 63.6 | 1.19 | 63.8 |
| BoN-GLASS (DDPM) | 38.6 | 70.8 | 23.8 | 71.5 | 0.316 | 68.8 | 1.33 | 71.8 |
| BoN-GLASS ($\rho = 0.4$) | 38.8 | 71.8 | 23.8 | 69.9 | 0.316 | 69.1 | 1.32 | 71.9 |
| FKS-GLASS (DDPM) | 39.7 | 70.5 | **24.1** | 68.8 | 0.317 | 68.3 | 1.37 | 68.2 |
| FKS-GLASS ($\rho = 0.4$) | **39.8** | **72.6** | **24.1** | **72.2** | **0.318** | **70.3** | **1.40** | **74.3** |

ODEs, i.e. the performance gain by using ODEs compared to SDEs weighs more than using SMC vs. Best-of-N. However, replacing the SDE transitions with GLASS Flows (our method), we remove this trade-off. In fact, **GLASS Flows combined with FKS leads to significant improvements for all 4 rewards models without sacrificing performance on GenEval**. We repeat the experiment on the PartiPrompts benchmark (Yu et al., 2022). Here, we optimize each reward model and evaluate on all other models. As shown in app. fig. 9, similarly FKS with **GLASS Flows leads to significant improvements and also constitutes the best-performing method on PartiPrompts**.

**Combining GLASS-FKS with reward guidance.** Finally, we explore combining FKS-GLASS with reward guidance. We pick the best-performing reward from table 1, ImageReward, and use it to compute the gradients in eq. (10) (see appendix D.4 for details). Note that we decrease the resolution of the image to $672 \times 672$ as the high-resolution images generated by FLUX led to memory bottlenecks in the gradient computation. We present results in table 2. As one can see, **guidance can improve results from FKS further - increasing both the reward being optimized and the GenEval results**. Finally, we note that we also explored reward guidance as a standalone method, showing GLASS Flows led to an improved trade-off between reward optimization and image quality. As reward guidance is less commonly used to improve text-to-image alignment as most improvements come from SMC (also in previous methods, e.g. (Singhal et al., 2025)), we present these results in appendix D.4.

Table 2: Improving GLASS-FKS using gradient guidance. ImageReward (IR) and GenEval results. Note: benchmarks are slightly different to table 1 as image resolution was decreased.

| Algorithm | IR | GenEv. |
|---|---|---|
| Flow baseline | 0.88 | 63.8 |
| FKS-GLASS | 1.45 | 72.7 |
| FKS-GLASS+$\nabla$ | **1.52** | **73.1** |

## 7 CONCLUSION

We introduce GLASS Flows, a novel way of sampling Markov transitions in flow and diffusion models using an "inner" flow matching model. This inner flow matching model can be retrieved from existing pre-trained models using sufficient statistics. Here, we applied GLASS Flows to inference-time reward alignment: Traditional sampling procedures for flow matching and diffusion models are poorly suited for inference-time reward alignment, either having deterministic trajectories like ODEs or requiring many steps to accurately simulate like SDEs. By combining the efficiency of ODEs with the stochasticity of SDEs, GLASS Flows substantially improve prior algorithms for inference-time reward alignment that relied on SDE sampling. Hence, GLASS Flows serve as a simple, drop-in solution for inference-time scaling of flow and diffusion models. In the future, one could explore applying GLASS Flows to other methods relying on SDE sampling, e.g. some reward fine-tuning (Liu et al., 2025; Xue et al., 2025; Li et al., 2025a; Domingo-Enrich et al., 2024) or image editing methods (Meng et al., 2021; Nie et al., 2023). Further, one could explore learning or dynamically adjusting the correlation parameter $\rho$ defining the GLASS transitions.

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

# A  PROOFS

## A.1  DETAILS ON FLOW MATCHING AND DIFFUSION BACKGROUND

**Vector field and denoiser.** We briefly present here the parameterizations between the vector field, denoiser, and score function. The conditional vector field $u_t(x|z)$ for Gaussian probability paths is given by (see (Lipman et al., 2022; 2024)):

$$u_t(x|z) = \frac{\dot{\sigma}_t}{\sigma_t}x + (\dot{\alpha}_t - \alpha_t\frac{\dot{\sigma}_t}{\sigma_t})z \tag{23}$$

Therefore, we know that

$$u_t(x) = \int u_t(x|z)p_{1|t}(z|x)\mathrm{d}z \tag{24}$$

$$= \frac{\dot{\sigma}_t}{\sigma_t}x + (\dot{\alpha}_t - \alpha_t\frac{\dot{\sigma}_t}{\sigma_t})\int zp_{1|t}(z|x)\mathrm{d}z \tag{25}$$

$$= \frac{\dot{\sigma}_t}{\sigma_t}x + (\dot{\alpha}_t - \alpha_t\frac{\dot{\sigma}_t}{\sigma_t})D_t(x) \tag{26}$$

Rearranging this equation results in:

$$D_t(x) = \int zp_{1|t}(z|x)\mathrm{d}z = \frac{1}{\dot{\alpha}_t\sigma_t - \alpha_t\dot{\sigma}_t}(\sigma_t u_t(x) - \dot{\sigma}_t x) \tag{27}$$

**Score function and probability flow ODE.** For completeness, we re-derive here the known connection between the flow matching ODE and the probability flow ODE in the score-based diffusion literature (Song et al., 2020b). We know that the score function is given by:

$$\nabla \log p_t(x|z) = \frac{\alpha_t z - x}{\sigma_t^2}$$

$$\Rightarrow \quad \nabla \log p_t(x) = \int \nabla \log p_t(x|z)p_{1|t}(z|x)\mathrm{d}z = \frac{\alpha_t D_t(x) - x}{\sigma_t^2}$$

$$\Rightarrow \quad D_t(x) = \frac{1}{\alpha_t}x + \frac{\sigma_t^2}{\alpha_t}\nabla \log p_t(x)$$

Therefore, plugging this into the denoiser-vector field identity (see eq. (26)) we get that

$$u_t(x) = \frac{\dot{\sigma}_t}{\sigma_t}x + (\dot{\alpha}_t - \alpha_t\frac{\dot{\sigma}_t}{\sigma_t})D_t(x) \tag{28}$$

$$= \frac{\dot{\alpha}_t}{\alpha_t}x + \left(\dot{\alpha}_t\frac{\sigma_t^2}{\alpha_t} - \dot{\sigma}_t\sigma_t\right)\nabla \log p_t(x) \tag{29}$$

$$= \frac{\dot{\alpha}_t}{\alpha_t}x + \frac{\nu_t^2}{2}\nabla \log p_t(x) \tag{30}$$

where $\nu_t^2 = 2\dot{\alpha}_t\sigma_t^2/\alpha_t - 2\dot{\sigma}_t\sigma_t$ as in eq. (5). Defining the forward drift function $\tilde{f}(x, t) = -\frac{\dot{\alpha}_{t-1}}{\alpha_{1-t}}x$ and the forward diffusion coefficient as $\tilde{\nu}_t = \nu_{1-t}$, then eq. (30) is the vector field of the probability flow ODE (Song et al., 2020b, Equation (13)) of the forward diffusion process given by (note that the diffusion literature uses a different time convention where $t = 0$ is $p_{\mathrm{data}}$ and $t \to \infty$ corresponds to noise):

$$\tilde{X}_0 \sim p_{\mathrm{data}}, \quad \mathrm{d}\tilde{X}_t = \tilde{f}(\tilde{X}_t, t)\mathrm{d}t + \tilde{\nu}_t\mathrm{d}\tilde{W}_t \tag{31}$$

where $\tilde{W}_t$ is a Brownian motion. Therefore, ODE sampling in eq. (4) is equivalent to the probability flow ODE for Gaussian probability paths (Song et al., 2020b, Equation (13)) (up to differing time convention).

**Time-reversal and DDPM.** Further, it is well known the SDE in eq. (31) has a time-reversal given by the SDE with diffusion coefficient $\nu_t$ and vector field given by (Anderson, 1982; Song et al., 2020b)

$$\mathrm{d}X_t = \left[\frac{\dot{\alpha}_t}{\alpha_t}x + \nu_t^2\nabla \log p_t(x)\right]\mathrm{d}t + \nu_t\mathrm{d}W_t \tag{32}$$

Using eq. (30), we can convert this back to the following form:

$$dX_t = \left[u_t(X_t) + \frac{1}{2}\nu_t^2 \nabla \log p_t(x)\right] dt + \nu_t dW_t \tag{33}$$

This proves eq. (5).

## A.2 Proof of Proposition 1

**Proposition 1.** For $\rho = \frac{\alpha_t \sigma_{t'}}{\sigma_t \alpha_{t'}}$, we get that: $p_{t'|t}^{\text{DDPM}}(X_{t'}|X_t) = p_{t'|t}(X_{t'}|X_t)$, i.e. DDPM transitions are a special case of GLASS transitions.

*Proof.* To show that

$$p_{t'|t}^{\text{DDPM}}(X_{t'}|X_t) = p_{t'|t}(X_{t'}|X_t)$$

it is sufficient to show that the joint distributions coincide

$$p_{t,t'}^{\text{DDPM}}(X_t, X_{t'}) = p_{t,t'}(X_t, X_{t'})$$

In turn, it is enough to show that the distribution conditioned on $z$ is the same

$$p_{t,t'}^{\text{DDPM}}(X_t, X_{t'}|z) = p_{t,t'}(X_t, X_{t'}|z) = \prod_{j=1}^{d} \mathcal{N}\left((X_t^j, X_{t'}^j); z^j\mu, \Sigma\right) \tag{34}$$

where we used eq. (12). For $\rho = \frac{\alpha_t}{\alpha_{t'}}\frac{\sigma_{t'}}{\sigma_t}$ as assumed, we obtain that

$$\mu = \begin{pmatrix} \alpha_t \\ \alpha_{t'} \end{pmatrix}, \quad \Sigma = \begin{pmatrix} \sigma_t^2 & \frac{\alpha_t}{\alpha_{t'}}\sigma_{t'}^2 \\ \frac{\alpha_t}{\alpha_{t'}}\sigma_{t'}^2 & \sigma_{t'}^2 \end{pmatrix} \tag{35}$$

In turn, as the DDPM is the time-reversal of a autoregressive forward process, it holds that for $t < t'$:

$$p_{t,t'}^{\text{DDPM}}(x_t, x_{t'}|z) = p_{t|t'}^{\text{DDPM}}(x_t|x_{t'})p_{t'}^{\text{DDPM}}(x_{t'}|z)$$

where we used that $p_{t|t'}^{\text{DDPM}}(x_t|x_{t'}, z) = p_{t|t'}^{\text{DDPM}}(x_t|x_{t'})$ as the DDPM process is also Markov in backwards time and it holds that

$$p_{t'}(x_{t'}|z) = \mathcal{N}(x_{t'}; \alpha_{t'}z, \sigma_{t'}^2 I_d)$$

$$p_{t|t'}^{\text{DDPM}}(x_t|x_{t'}) = \mathcal{N}\left(x_t; \frac{\alpha_t}{\alpha_{t'}}x_{t'}, (\sigma_t^2 - \frac{\alpha_t^2}{\alpha_{t'}^2}\sigma_{t'}^2)I_d\right)$$

where the second equation follows from the fact $p_{t'|t}^{\text{DDPM}}$ is the transition kernel of the forward noising process (see eq. (31)), which is a Gaussian Markov process in discrete time (which is unique if we restrict to have marginals given by $p_t$). Alternatively, one can also directly prove this by using that $p_{t|t'}^{\text{DDPM}}$ is the transition kernel of the forward process in eq. (31) and using the transition kernels of Ohrnstein-Uhlenbeck processes, see e.g. (Karras et al., 2022, equation (11))).

It remains to work out the mean and covariance of the joint distribution using classical rules for Gaussian distributions. Specifically, we can sample from $p_{t,t'}^{\text{DDPM}}(\cdot|z)$ by first sampling $X_{t'} \sim p_{t'}(x_{t'}|z)$ and then sampling

$$X_t = \frac{\alpha_t}{\alpha_{t'}}X_{t'} + \sqrt{\sigma_t^2 - \frac{\alpha_t^2}{\alpha_{t'}^2}\sigma_{t'}^2}\epsilon, \quad \epsilon \sim \mathcal{N}(0, I_d)$$

Therefore, it holds that the conditional means are given by

$$\mathbb{E}[X_{t'}|Z=z] = \alpha_{t'}z, \quad \mathbb{E}[X_t|Z=z] = \frac{\alpha_t}{\alpha_{t'}}\mathbb{E}[X_{t'}|Z=z] = \frac{\alpha_t}{\alpha_{t'}}\alpha_{t'}z = \alpha_t z$$

and

$$\text{Cov}[X_t, X_{t'}|Z=z] = \frac{\alpha_t}{\alpha_{t'}}\text{Cov}[X_{t'}, X_{t'}|Z=z] + \sqrt{\sigma_t^2 - \frac{\alpha_t^2}{\alpha_{t'}^2}\sigma_{t'}^2}\underbrace{\text{Cov}[\epsilon, X_{t'}|Z=z]}_{=0}$$

$$= \frac{\alpha_t}{\alpha_{t'}}\sigma_{t'}^2$$

$$\text{Var}[X_{t'}|Z=z] = \sigma_{t'}^2, \quad \text{Var}[X_t|Z=z] = \frac{\alpha_t^2}{\alpha_{t'}^2}\sigma_{t'}^2 + (\sigma_t^2 - \frac{\alpha_t^2}{\alpha_{t'}^2}\sigma_{t'}^2) = \sigma_t^2$$

Therefore, we see that eq. (34) holds. This finishes the proof. $\square$

## A.3 PROOF OF PROPOSITION 2

We start by making a statement about summarizing 2 Gaussian measurements into 1 measurement.

**Lemma 1** (Equivalent observations for multivariate Gaussian). Let $z \in \mathbb{R}$, let $\mu = (\mu_1, \mu_2)^T \in \mathbb{R}^2$ be a mean vector and $\Sigma \in \mathbb{R}^{2 \times 2}$ a (positive definite) covariance matrix. Further, let $\mathbf{X} \in \mathbb{R}^2$ be a multivariate Gaussian random variable given by

$$\mathbf{X} = (X_1, X_2)^T \sim \mathcal{N}(z\mu, \Sigma)$$

Further, define the one-dimensional random variable $Y$ via the mapping

$$Y = S(\mathbf{X}) \quad \text{where } S(\mathbf{x}) = \frac{\mu^T \Sigma^{-1} \mathbf{x}}{\mu^T \Sigma^{-1} \mu}, \quad \mathbf{x} = (x_1, x_2)^T \in \mathbb{R}^2 \tag{36}$$

Then observing $\mathbf{X}$ at value $\mathbf{X} = \mathbf{x}$ is equivalent to observing $Y$ at $Y = S(\mathbf{x})$, i.e. for any prior distribution $p_{\text{data}}$ of $Z = z$ it holds that

$$\underbrace{p(Z|\mathbf{X} = \mathbf{x})}_{\text{posterior with observation } \mathbf{X}} = \underbrace{p(Z|Y = S(\mathbf{x}))}_{\text{posterior with observation } Y} \tag{37}$$

Equivalently, $S(\mathbf{x})$ is a sufficient statistic for $\mathbf{X}$ given $z$. Further, $Y$ is again normally distributed with

$$Y \sim \mathcal{N}\left(z, \frac{1}{\mu^T \Sigma^{-1} \mu}\right) \tag{38}$$

i.e. observing $X_1, X_2$ is equivalent to observing a single Gaussian measurement of $z$.

*Proof.* The fact that $Y$ is again normally distributed follows from the fact that linear mappings of Gaussians are again Gaussian with mean and variances given by known formulas:

$$\mathbb{E}[Y] = \frac{1}{\mu^T \Sigma^{-1} \mu} \mathbb{E}[\mu^T \Sigma^{-1} \mathbf{X}] = \frac{1}{\mu^T \Sigma^{-1} \mu} \mu^T \Sigma^{-1} \mathbb{E}[\mathbf{X}] = \frac{1}{\mu^T \Sigma^{-1} \mu} \mu^T \Sigma^{-1} \mu z = z$$

$$\mathbb{V}[Y] = \frac{1}{(\mu^T \Sigma^{-1} \mu)^2} (\mu^T \Sigma^{-1}) \Sigma (\mu^T \Sigma^{-1})^T = \frac{1}{\mu^T \Sigma^{-1} \mu}$$

Further, it holds that

$$\log p(\mathbf{X}|Z = z)$$
$$= -\frac{1}{2}(\mathbf{X} - z \cdot \mu)^T \Sigma^{-1}(\mathbf{X} - z \cdot \mu)$$
$$= C(\mathbf{X}) + \frac{1}{2} z \mu^T \Sigma^{-1} \mathbf{X} + \frac{1}{2} \mathbf{X}^T \Sigma^{-1} \mu z - \frac{1}{2} z^2 \mu^T \Sigma^{-1} \mu$$
$$= C(\mathbf{X}) + \frac{\mu^T \Sigma^{-1} \mu}{2} \left[ z S(\mathbf{X}) + S(\mathbf{X}) z - z^2 \right]$$
$$= C(\mathbf{X}) - \frac{(z - S(\mathbf{X}))^2}{2(\mu^T \Sigma^{-1} \mu)^{-1}}$$
$$= C(\mathbf{X}) + \log p(Y = S(\mathbf{X})|Z = z)$$

where $C(\mathbf{X})$ is an arbitrary constant independent of $z$. Therefore,

$$p(\mathbf{X}|Z = z) = \exp(C(\mathbf{X}))p(Y = S(\mathbf{X})|Z = z)$$

Hence, we know that

$$p(Z|\mathbf{X} = \mathbf{x}) \propto p(Z)p(\mathbf{X} = \mathbf{x}|Z) \propto p(Z)p(Y = S(\mathbf{X})|Z) \propto p(Z|Y = S(\mathbf{X}))$$

where we dropped constants in $Z$. As both sides are distributions in $Z$ (i.e. integrate to 1), they must be equal. This finishes the proof. $\square$

**Proposition 2.** Let $\mathbf{x} = (x_1, x_2)$ with $\mathbf{x}_i \in \mathbb{R}^d$ and $t^* = t^*(\mu, \Sigma) = g^{-1}((\mu\Sigma^{-1}\mu)^{-1})$. Then:

$$\underbrace{D_{\mu,\Sigma}(\mathbf{x})}_{\text{GLASS denoiser}} = \underbrace{D_{t^*}(\alpha_{t^*}S(\mathbf{x}))}_{\text{``standard'' pre-trained denoiser with reparameterized input and time}}$$

where $D_t$ is defined as in eq. (14) and $\alpha_t$ is the scheduler in eq. (2).

*Proof.* By lemma 1, we know that

$$D_{\mu,\Sigma}(\mathbf{x}) = \int zp(Z = z|\mathbf{X} = \mathbf{x})\mathrm{d}z = \int zp(Z = z|Y = S(\mathbf{x}))\mathrm{d}z = \int zp(Z = z|\alpha_t Y = \alpha_t S(\mathbf{x}))\mathrm{d}z$$

for any $0 < t \leq 1$. We know that $\alpha_t Y$ given $Z = z$ has distribution

$$\alpha_t Y \sim \mathcal{N}\left(\alpha_t z, \frac{\alpha_t^2}{\mu^T \Sigma^{-1}\mu}\right)$$

Now the right-hand side, we want to coincide with a time point $t$ in the Gaussian probability path, i.e. such that

$$\mathcal{N}\left(\alpha_t z, \frac{\alpha_t^2}{\mu^T \Sigma^{-1}\mu}\right) = \mathcal{N}(\alpha_t z; \sigma_t^2 I_d)$$

This is equivalent to

$$g(t) = \frac{\sigma_t^2}{\alpha_t^2} = \frac{1}{\mu^T \Sigma^{-1}\mu}$$

Now, by assumption $\alpha_t$ is strictly monotonically increasing and $\sigma_t$ is strictly monotonically decreasing. Therefore, the function $g$ is invertible and we can simply set $t^*$ accordingly as stated in theorem. Then, we get:

$$D_{\mu,\Sigma}(\mathbf{x}) = \int zp(Z = z|X_t = \alpha_{t^*}S(\mathbf{x}))\mathrm{d}z = D_{t^*}(\alpha_{t^*}S(\mathbf{x}))$$

$\square$

### A.4 PROOF OF THEOREM 1

**Theorem 1.** Let us be given two times $t < t'$, a starting point $x_t$, and a correlation parameter $\rho$ defining the GLASS transition $p_{t'|t}$ in eq. (13). Then we can sample from $p_{t'|t}(\cdot|x_t)$ as follows:

Define the **GLASS velocity field** as the weighted sum of $\bar{x}_s, x_t$ and the GLASS denoiser

$$u_s(\bar{x}_s|x_t, t) = w_1(s)\bar{x}_s + w_2(s)D_{\mu(s),\Sigma(s)}(x_t, \bar{x}_s) + w_3(s)x_t \tag{19}$$

with weight coefficients $w_1(s), w_2(s), w_3(s) \in \mathbb{R}$ and time-dependent mean scale and covariance $\mu(s), \Sigma(s)$ given by

$$\mu(s) = \left(\frac{\alpha_t}{\bar{\alpha}_s + \bar{\gamma}\alpha_t}\right), \quad \Sigma(s) = \begin{pmatrix} \sigma_t^2 & \sigma_t^2\bar{\gamma} \\ \sigma_t^2\bar{\gamma} & \bar{\sigma}_s^2 + \bar{\gamma}^2\sigma_t^2 \end{pmatrix} \tag{20}$$

$$w_1(s) = \frac{\partial_s\bar{\sigma}_s}{\bar{\sigma}_s}, \quad w_2(s) = \partial_s\bar{\alpha}_s - \bar{\alpha}_s w_1(s), \quad w_3(s) = -\bar{\gamma}w_1(s) \tag{21}$$

where $\bar{\alpha}_s, \bar{\sigma}_s, \bar{\gamma}$ are chosen as in eq. (18). Then the final point $\bar{X}_1$ of the trajectory $\bar{X}_s$ obtained via the ODE

$$\bar{X}_0 \sim \mathcal{N}(\bar{\gamma}x_t, \bar{\sigma}_0^2 I_d), \quad \frac{\mathrm{d}}{\mathrm{d}s}\bar{X}_s = u_s(\bar{X}_s|x_t, t) \tag{22}$$

is a sample from the GLASS transition, i.e. $\bar{X}_1 \sim p_{t'|t}(\cdot|x_t)$. More generally, $\bar{X}_s \sim p_s(\cdot|x_t)$ for all $0 \leq s \leq 1$.

*Proof.* We can obtain samples $\bar{X}_s$ from the probability path $p_s(\bar{X}_s|X_t, t, z) = \mathcal{N}(\bar{x}_s; \bar{\alpha}_s z + \bar{\gamma}x_t, \bar{\sigma}_s^2 I_d)$ by

$$\bar{X}_s = \bar{\alpha}_s Z + \bar{\gamma} X_t + \bar{\sigma}_s \epsilon, \quad \epsilon \sim \mathcal{N}(0, I), Z \sim p_{\text{data}} \tag{39}$$

Therefore, the derivative with respect to $s$ is given by

$$\partial_s \bar{X}_s = \partial_s \bar{\alpha}_s Z + \partial_s \bar{\sigma}_s \epsilon \tag{40}$$

Now, we can reparameterize $\epsilon$ into $X_t$ and $\bar{X}_s$

$$\epsilon = \frac{1}{\bar{\sigma}_s} \left[ \bar{X}_s - \bar{\alpha}_s Z - \bar{\gamma} X_t \right]$$

Inserting this into eq. (40), we get

$$\partial_s \bar{X}_s = \partial_s \bar{\alpha}_s Z + \partial_s \bar{\sigma}_s \frac{1}{\bar{\sigma}_s} \left[ \bar{X}_s - \bar{\alpha}_s Z - \bar{\gamma} X_t \right]$$
$$= w_1(s) \bar{X}_s + w_2(s) Z + w_3(s) X_t$$

where $w_1, w_2, w_3$ are as in eq. (21). Taking the conditional expectation, we get:

$$\mathbb{E}[\partial_s \bar{X}_s | X_t = x_t, \bar{X}_s = \bar{x}_s]$$
$$= w_1(s) \bar{x}_s + w_2(s) \mathbb{E}[Z | X_t, \bar{X}_s] + w_3(s) x_t$$
$$= w_1(s) \bar{x}_s + w_2(s) D_{\mu(s), \Sigma(s)}(x_t, \bar{x}_s) + w_3(s) x_t$$
$$= u_s(\bar{x}_s | x_t, t)$$

where $u_s(\bar{x}_s | x_t, t)$ is defined as in the theorem. It remains to show that the left-hand side of the equation fulfills the continuity for the probability path $p_s(\bar{x}_s | x_t, t)$. Let $f : \mathbb{R}^d \to \mathbb{R}$ be an arbitrary smooth function with compact support (test function). Then we have

$$\int f(\bar{x}) \partial_s p_s(\bar{x}|x_t, t) d\bar{x}$$
$$= \partial_s \int f(\bar{x}) p_s(\bar{x}|x_t, t) d\bar{x}$$
$$= \partial_s \mathbb{E}[f(\bar{X}_s) | X_t = x_t]$$
$$= \mathbb{E}[\nabla f(\bar{X}_s)^T \partial_s \bar{X}_s | X_t = x_t]$$
$$= \mathbb{E}[\nabla f(\bar{X}_s)^T \mathbb{E}[\partial_s \bar{X}_s | \bar{X}_s, X_t] | X_t = x_t]$$
$$= \mathbb{E}[\nabla f(\bar{X}_s)^T u_s(\bar{X}_s | X_t, t) | X_t = x_t]$$
$$= \int \nabla f(\bar{x})^T u_s(\bar{x}|x_t, t) p_s(\bar{x}|x_t, t) d\bar{x}$$
$$= \int f(\bar{x}) \left[ -\nabla_{\bar{x}} \cdot (u_s(\bar{x}|x_t, t) p_s(\bar{x}|x_t, t)) \right] d\bar{x}$$

where we used partial integration in the last step. As $f$ is an arbitrary test function, we obtain that both sides also coincide for each point:

$$\partial_s p_s(\bar{x}|x_t, t) = -\nabla_{\bar{x}} \cdot (u_s(\bar{x}|x_t, t) p_s(\bar{x}|x_t, t))$$

This shows that the continuity equation is fulfilled (see e.g. (Lipman et al., 2022; 2024)). This implies that the trajectory $\bar{X}_s$ obtained via the ODE

$$\bar{X}_0 \sim \mathcal{N}(\bar{\gamma} x_t, \bar{\sigma}_0^2 I_d), \quad \frac{d}{ds} \bar{X}_s = u_s(\bar{X}_s | x_t, t) \tag{41}$$

has a final point $\bar{X}_1$ that is a sample from the GLASS transition:

$$\bar{X}_s \sim p_s(\cdot | X_t = x_t) \tag{42}$$

for all $0 \leq s \leq 1$. This finishes the proof. $\qquad\square$

## B  ADDITIONAL GLASS DISCUSSION

### B.1  GLASS FLOWS AND REWARD GUIDANCE

We explain how (gradient) guidance aimed at sampling from the reward-tilted distribution as defined in Section 3 can be simply applied to GLASS Flows. We first recall the construction for "standard" FM and diffusion models and then show how to translate it to guidance for GLASS Flows.

**Guidance for "standard" FM models.**  Recall that to sample from the reward-tilted distribution $p^r(x)$ in our setting, the tilted vector field $u_t^r(x)$ can be written in terms of marginal vector field $u_t(x)$ and the value function $V_t(x)$

$$u_t^r(x) = u_t(x) + c_t \nabla V_t(x), \tag{43}$$

where $c_t = \frac{\dot{\alpha}_t}{\alpha_t}\sigma_t^2 - \dot{\sigma}_t\sigma_t$. Equivalently, in the denoiser parameterization

$$D_t^r(x) = D_t(x) + \frac{\sigma_t^2}{\alpha_t}\nabla V_t(x). \tag{44}$$

In practice, $V_t(x)$ and therefore $D_t^r(x)$ is often approximated via (Chung et al., 2022)

$$V_t(x) \approx \beta_t r(D_t(x)) \tag{45}$$

where $r_t$ as in eq. (10) and $\beta_t \geq 0$ is a hyperparameter (theoretically, $\beta_t = 1$ would be ideal, it is common to tune this hyperparameter however). Therefore, the final approximated guidance vector is given by

$$u_t^r(x) = u_t(x) + c_t\beta_t\nabla_x[r(D_t(x))]$$

**Guidance for GLASS Flows.**  To derive guidance for GLASS Flows, we now translate the same principles to GLASS Flows. For this, let $D_{\mu,\Sigma}^r(\mathbf{x})$ be the denoiser for the reward-tilted distribution. Then we know that:

$$D_{\mu(s),\Sigma(s)}^r(\mathbf{x}) = D_{t^*}^r\left(\alpha_{t^*}S(\mathbf{x})\right) \tag{46}$$

Further, using the same approximation in eq. (45) and inserting it into eq. (44), we get:

$$\begin{aligned}
D_{\mu(s),\Sigma(s)}^r(\mathbf{x}) &= D_{t^*}^r\left(\alpha_{t^*}S(\mathbf{x})\right) \\
&\approx D_{t^*}\left(\alpha_{t^*}S(\mathbf{x})\right) + \beta_{t^*}\frac{\sigma_{t^*}^2}{\alpha_{t^*}}\nabla_y r(D_{t^*}(y))_{|y=\alpha_{t^*}S(\mathbf{x})} \\
&= D_{\mu(s),\Sigma(s)}(\mathbf{x}) + \beta_{t^*}\frac{\sigma_{t^*}^2}{\alpha_{t^*}}\nabla_y r(D_{t^*}(y))_{|y=\alpha_{t^*}S(\mathbf{x})}
\end{aligned}$$

Finally, we can insert this identity into the formula for the tilted GLASS velocity field (see theorem 1):

$$u_s^r(\bar{x}_s|x_t, t) \tag{47}$$
$$= w_1(s)\bar{x}_s + w_2(s)D_{\mu(s),\Sigma(s)}^r(x_t, \bar{x}_s) + w_3(s)x_t \tag{48}$$

$$= \underbrace{w_1(s)\bar{x}_s + w_2(s)D_{\mu(s),\Sigma(s)}(x_t, \bar{x}_s) + w_3(s)x_t}_{=u_s(\bar{x}_s|x_t, t)} + \beta_{t^*}\frac{\sigma_{t^*}^2}{\alpha_{t^*}}w_2(s)\nabla_y r(D_{t^*}(y))_{|y=\alpha_{t^*}S(\mathbf{x})} \tag{49}$$

$$= u_s(\bar{x}_s|x_t, t) + \beta_{t^*}\frac{\sigma_{t^*}^2}{\alpha_{t^*}}w_2(s)\nabla_y r(D_{t^*}(y))_{|y=\alpha_{t^*}S(\mathbf{x})} \tag{50}$$

where $u_s(\bar{x}_s|x_t, t)$ is the GLASS velocity field for the (non-tilted) distribution $p_{\text{data}}$. Theoretically, $\beta_t = 1$ would be optimal for a perfect estimation of the value function. However, because of the approximation in eq. (45), we recommend tuning $\beta_t \geq 0$ as done already for previous guidance methods (Chung et al., 2022; He et al., 2023).

## B.2 $M = 1$ GLASS FLOWS

As described in Section 4.2.2, GLASS Flows generate a data point using $K$ transitions and $M$ simulation steps per transition. For $K = 1$, GLASS Flows are equal to standard flow matching integration of an ODE performed over $M$ simulation steps. In this section, we instead consider when $M = 1$. Let $\epsilon \sim \mathcal{N}(0, I_d)$, where $\bar{x}_0 = \bar{\gamma}x_t + \bar{\sigma}_0\epsilon$. Then for one-step integration using the CondOT schedulers, we get

$$
\begin{aligned}
\bar{x}_1 &= \bar{x}_0 + u_0(\bar{x}_0|x_t, t) \\
&= \bar{x}_0 + w_1(0)\bar{x}_0 + w_2(0)D_{\mu(0),\Sigma(0)}(x_t, \bar{x}_0) + w_3(0)x_t \\
&= \bar{x}_0 + w_1(0)\bar{x}_0 + w_2(0)D_t(x_t) + w_3(0)x_t \\
&= \bar{x}_0 + w_1(0)\bar{x}_0 + w_2(0)D_t(x_t) + w_3(0)x_t \\
&= \bar{x}_0 + \frac{\bar{\sigma} - \bar{\sigma}_0}{\bar{\sigma}_0}\bar{x}_0 + \bar{\alpha}D_t(x_t) - \bar{\gamma}\frac{\bar{\sigma} - \bar{\sigma}_0}{\bar{\sigma}_0}x_t \\
&= \bar{\gamma}x_t + \bar{\alpha}D_t(x_t) + \bar{\sigma}\epsilon.
\end{aligned}
\tag{51}
$$

Comparing with the conditional Gaussian probability path evaluated at $s = 1$, $p_1(\bar{X}_1 = x|x_t, z) = p_{t'|t}(\bar{X}_{t'} = x|x_t, z)$, we note that GLASS Flows for $M = 1$ samples transitions via

$$
\bar{X}_{t'} \sim p_{t'|t}(\bar{X}_{t'}|x_t, z = D_t(x_t)).
\tag{52}
$$

This is identical to the Gaussian transition kernel parameterization typically used in discrete-time diffusion models. So at $M = 1$, transitions from GLASS Flows match a discrete-time diffusion model parameterized in this fashion with the same Gaussian kernel and denoiser.

In fact, $M = 1$ GLASS Flows are exactly equal to denoising-diffusion implicit models (DDIM) (Song et al., 2020a) for particular GLASS parameters and the same pre-trained denoiser. We begin by noting that DDIM uses a model parameterization that inserts the denoiser for $z$. Next, we demonstrate that the $z$-conditional transition kernels are equal for particular GLASS parameters. DDIM uses a conditional parameter per transition from $t$ to $t'$, $\sigma^D_{t',t}$, and marginal parameters $\alpha^D_t$, where $0 \le \alpha^D_t \le 1$ and $0 \le (\sigma^D_{t',t})^2 \le 1 - \alpha^D_{t'}$, and superscript $D$ denotes DDIM. From Equation 16, an arbitrary GLASS transition kernel can be written

$$
p_{t'|t}(x_{t'}|x_t, z) = \mathcal{N}(\alpha_{t'}z + \rho_{t',t}\frac{\sigma'_t}{\sigma_t}(x_t - \alpha_t z), \sigma^2_{t'}(1 - \rho^2_{t',t})I),
\tag{53}
$$

where $0 \le \rho^2_{t',t} \le 1$ and $\rho$'s explicit dependence on $t$ and $t'$ is included for clarity. Now set

$$
\begin{aligned}
\alpha_t &= \sqrt{\alpha^D_t} \\
\sigma^2_t &= 1 - \alpha^D_t \\
\rho_{t,t'} &= \sqrt{1 - \frac{(\sigma^D_{t,t'})^2}{1 - \alpha^D_{t'}}},
\end{aligned}
\tag{54}
$$

where we note that $0 \le \rho^2_{t',t} \le 1$ is satisfied due to the constraint on $\sigma^D_{t,t'}$. Inserting and rearranging, we recover the DDIM transition kernel (see Eq. 7 in Song et al. (2020a)).

$$
p^{DDIM}_{t'|t}(x_{t'}|x_t, z) = \mathcal{N}(\sqrt{\alpha^D_{t'}}z + \sqrt{1 - \alpha^D_{t'} - (\sigma^D_{t',t})^2}\frac{\left(x_t - \sqrt{\alpha^D_t}z\right)}{\sqrt{1 - \alpha^D_t}}, (\sigma^D_{t',t})^2 I).
\tag{55}
$$

## B.3 NUMERICAL STABILITY

In the following, we show that is simple to ensure numerical stability in algorithm 1. Generally, we recommend performing all operations in algorithm 1 - except the neural network evaluation - in higher precision (float64). This has minimal overhead compared to large-scale neural network evaluations and minimize errors from reparameterization. We now discuss more specific steps.

$s = 0$ **edge case.** For $s = 0$, it holds that

$$\bar{X}_0 = \bar{\gamma} X_t + \bar{\sigma}_0 \epsilon \tag{56}$$

One could think of this as a 2-step Markov chain, i.e. it holds $p(\bar{X}_0|X_t, z) = p(\bar{X}_0|X_t)$. It is a classical property of Markov chains that the posterior then also depends only on the state $X_t$:

$$p(z|X_t, \bar{X}_0) = p(z|X_t)$$

One can prove this directly by applying Bayes' rule twice and using the Markov property:

$$p(z|X_t, \bar{X}_0) \propto p(X_t, \bar{X}_0|z)p(z) \tag{57}$$
$$= p(\bar{X}_0|X_t, z)p(X_t|z)p(z) \tag{58}$$
$$= p(\bar{X}_0|X_t)p(X_t|z)p(z) \tag{59}$$
$$\propto p(X_t|z)p(z) \tag{60}$$
$$\propto p(z|X_t) \tag{61}$$

As the first and last term both integrate to 1 (as they are probability distributions over $z$), they must be equal. As the posteriors are the same, also the denoisers are the same:

$$D_{\mu(0), \Sigma(0)}(X_t, \bar{X}_0) = D_t(X_t)$$

In algorithm 1, we use this fact to ensure numerical stability at $s = 0$.

**Numerical stability of matrix inversion for $\Sigma(s)$ and weight coefficients.** The covariance matrix is given as:

$$\Sigma(s) = \begin{bmatrix} \sigma_t^2 & \sigma_t^2 \bar{\gamma} \\ \sigma_t^2 \bar{\gamma} & \bar{\sigma}_s^2 + \bar{\gamma}^2 \sigma_t^2 \end{bmatrix} = \begin{bmatrix} \sigma_t^2 & \sigma_{t'} \sigma_t \rho \\ \sigma_{t'} \sigma_t \rho & \bar{\sigma}_s^2 + \rho^2 \sigma_{t'}^2 \end{bmatrix} \tag{62}$$

where we inserted $\bar{\gamma} = \rho \sigma_{t'}/\sigma_t$. Then

$$\det \Sigma(s) = (\bar{\sigma}_s^2 + \rho^2 \sigma_{t'}^2)\sigma_t^2 - \sigma_{t'}^2 \sigma_t^2 \rho^2 = \sigma_t^2 \bar{\sigma}_s^2$$

Therefore, $\det \Sigma(s) > 0$ and $\Sigma(s)$ is invertible whenever $\sigma_t^2 > 0$ and $\bar{\sigma}_s^2 > 0$. We now discuss when this might *not* be the case. First, $\sigma_t^2 > 0$ is equivalent to $t < 1$ as $\sigma_t$ is strictly monotonically decreasing with $\sigma_1 = 0$ by assumption. Hence, $\sigma_t > 0$ always holds in practice as we never take a transition starting at the final time $t = 1$. However, it is important that the operation would not be well-defined in this case. Second, $\bar{\sigma}_s$ is also positive for $s < 1$ by assumption and it fulfills at $s = 1$ that $\bar{\sigma}_1^2 = \sigma_{t'}^2(1 - \rho^2)$. Therefore, for either $t' = 1$ or $\rho = \pm 1$, it would hold that $\bar{\sigma}_1 = 0$. Hence, for $t' = 1$ or $\rho = \pm 1$, the matrix $\Sigma(s)$ would not be invertible. However, in algorithm 1, we always simulate and take velocities for $s < 1$. Therefore, everything is well-defined and we observed that the inversion of $\Sigma(s)$ did not constitute a numerical problem even for $\rho = \pm 1$. In fact, even for $\rho = \pm 1$, the samples we obtain are of high quality (see experiments in appendix D.3). Further, one can add a small value to the diagonal matrix to make it invertible: $\Sigma(s) \leftarrow \Sigma(s) + \epsilon I_2$ for $\epsilon > 0$ to account for $s$ close to 1. Similarly, the weight coefficients for the GLASS velocity field are given by:

$$w_1(s) = \frac{\partial_s \bar{\sigma}_s}{\bar{\sigma}_s}, \quad w_2(s) = \partial_s \bar{\alpha}_s - \bar{\alpha}_s w_1(s), \quad w_3(s) = -\bar{\gamma} w_1(s) \tag{63}$$

for $\bar{\gamma} = \rho \sigma_{t'}/\sigma_t$. We sample transitions for $t < t' \leq 1$. Therefore, $t < 1$ and also $\sigma_t > 0$ and therefore $\bar{\gamma}$ is well-defined. Further, for $s = 1$ and $\rho = \pm 1$ or $t' = 1$, it holds that $w_1(s)$ is not well-defined as $\bar{\sigma}_s = 0$. However, as before, algorithm 1 only uses time steps $s \leq 1 - 1/M$ and therefore we did not encounter any numerical instabilities. As mentioned above, we recommend performing all operations in algorithm 1 - except the neural network evaluation - in higher precision (float64). This will have negligible overhead compared to neural network calls.

## B.4 OTHER DIFFUSION PARAMETERIZATIONS

**Other vector field parameterizations.** In algorithm 1, we assume that the pre-trained flow or diffusion model is given in the velocity parameterization $u_t(x)$ as used in flow matching. It is well-known that diffusion models can be equivalently parameterized via the score function $\nabla \log p_t(x)$ or the denoiser $D_t(x)$ or the noise predictor (also called $\epsilon$-predictor):

$$\epsilon_t(x_t) = \mathbb{E}[X_0|X_t = x_t]$$

To use a model trained with a different parameterization for GLASS Flows, we simply reparameterize them into the denoiser parameterization $D_t(x)$. In algorithm 1, we have presented this for the reparameterization of $u_t$ into $D_t$ - for other models one might simply have to use the corresponding reparameterization. See e.g. (Lipman et al., 2024, Table 1) for reparameterization formulas.

**Discrete-time parameterizations.** Our derivations assume that we have a model $u_t(x)$ trained in continuous time $0 \leq t \leq 1$. In contrast, discrete-time diffusion models (Sohl-Dickstein et al., 2015; Ho et al., 2020) are trained with a different time reparameterization. We discuss how to use GLASS Flows for these models. Let us assume that the discrete-time diffusion models is given in the shape of a denoiser model $\tilde{D}_k(x)$ with discrete time steps $k = 1, \cdots, N$ and discrete-time schedulers $\tilde{\alpha}_k, \tilde{\sigma}_k$. We can map these discrete-time $k = 1, \cdots, N$ into a grid $\mathcal{G} = \{\tilde{t}_j\}_{j=1,\cdots,N}$ of continuous time points $0 = \tilde{t}_1 < \tilde{t}_2 < \cdots < \tilde{t}_N = 1$ via

$$\tilde{t}_k = g^{-1}\left(\frac{\tilde{\sigma}_k^2}{\tilde{\alpha}_k^2}\right)$$

where $g(t) = \sigma_t^2/\alpha_t^2$ as before for continuous-time schedulers $\alpha_t, \sigma_t$. Further, define the denoiser model $D_t$ on the grid points as

$$D_{\tilde{t}_k}(x) = \tilde{D}_k\left(\frac{\tilde{\alpha}_k}{\alpha_t}x\right)$$

Note that $D_t(x)$ is a valid denoiser model for $t \in \mathcal{G}$ in the grid and schedulers $\alpha_t, \sigma_t$ - same as before. However, there is one important difference: querying $D_t(x)$ for $t \notin \mathcal{G}$ would correspond to an invalid input to $\tilde{D}_k$ or, at least, and out-of-domain query of the neural network. Naturally, we want to avoid such out-of-domain queries. Specifically, during simulation of the GLASS Flow for a transition from $t$ to $t'$, the denoiser model is queried at times $t^*(\mu(s), \Sigma(s))$ given by

$$t^*(\mu(s), \Sigma(s)) = g^{-1}((\mu(s)\Sigma^{-1}(s)\mu(s))^{-1})$$

$$\mu(s) = \begin{pmatrix} \alpha_t \\ \bar{\alpha}_s + \bar{\gamma}\alpha_t \end{pmatrix}, \quad \Sigma(s) = \begin{pmatrix} \sigma_t^2 & \sigma_t^2\bar{\gamma} \\ \sigma_t^2\bar{\gamma} & \bar{\sigma}_s^2 + \bar{\gamma}^2\sigma_t^2 \end{pmatrix}$$

It holds that $t^*(\mu(0), \Sigma(0)) = t$ (see appendix B.3) and therefore in particular we restrict transitions to only appear from grid points to grid points, i.e. $t, t' \in \mathcal{G}$. To choose "inner" grid points $0 = s_0 < s_1 < \cdots < s_M = 1$, we can restrict ourselves to the set $\mathcal{T} = \{s \in [0,1] | t^*(\mu(s), \Sigma(s)) \in \mathcal{G}\}$, i.e. choosing $s_i \in \mathcal{T}$. In general, there is no closed-form for $\mathcal{T}$ as we allow for general schedulers $\alpha_t, \sigma_t, \bar{\sigma}_s, \bar{\alpha}_s$. A simple numerical approach to (approximately) obtain $\mathcal{T}$ is always valid. If $\alpha_t, \sigma_t, \bar{\sigma}_s, \bar{\alpha}_s$ have simple analytical formulas, we might also obtain a closed form for $\mathcal{T}$. Therefore, GLASS Flows can be applied to discrete-time diffusion models in the same way and all of our results equally hold - with the only difference that the time points $s$ in algorithm 1 are constrained to $s \in \mathcal{T}$. Note that the above procedure is **not an approximation** but is exact: Simulating GLASS Flows with a reparameterized discrete-time diffusion model or a continuous-time diffusion model leads to identical results as long as time steps $s \in \mathcal{T}$ (assuming both models have no training error). Of course, there is still an error in the discretization of the simulation of the ODE - which would be identical for both, however. Therefore, GLASS Flows can also be applied to discrete-time diffusion models in the same way with a constrained set of valid inner grid points $s$ in algorithm 1.

## B.5 ROLE OF $\rho$

We briefly discuss the role of $\rho$ and how it determines the stochastic nature of the GLASS transition $p_{t'|t}(x_{t'}|x_t)$. Now that the GLASS transition can be written as:

$$p_{t'|t}(x_{t'}|x_t) = \int \underbrace{p_{t'|t}(x_{t'}|x_t, z)}_{\text{GLASS (depends on }\rho)} \underbrace{p_{1|t}(z|x_t)}_{\text{posterior (indep. of }\rho)} \, \mathrm{d}z$$

Note note that for $\rho = 1$, the conditional transition $p_{t'|t}(x_{t'}|x_t, z)$ becomes deterministic. However, $p_{1|t}(z|x_t)$ is independent of $\rho$ and is not deterministic but rather a proper posterior distribution. Therefore, even for $\rho = 1$, the GLASS transitions are stochastic (in particular, $\rho = 1$ does *not* correspond to ODE sampling). Therefore, $\rho$ determines $p_{t'|t}(x_{t'}|x_t, z)$, in particular how correlated is that we add to a single data point but only partially determines the probabilistic/stochastic transition. We found that the time difference $t' - t$ (determined by the number of transitions $K$) and the variance of the posterior $p_{1|t}$ are equally important factors determining the variance of the transition distribution $p_{t'|t}(x_{t'}|x_t)$.

### B.6 ANALYTICAL FORMULAS FOR $g^{-1}$

We derive specific formulas for $g, g^{-1}$ for various schedulers $\alpha_t, \sigma_t$. The inversion $g^{-1}$ is used in algorithm 1. Recall the definition:

$$g(t) = \frac{\sigma_t^2}{\alpha_t^2}$$

For choices of $\alpha_t, \sigma_t$, one can derive $g^{-1}$ analytically. We present the derivation for the most common schedulers.

**Linear schedule.** Let us set

$$\alpha_t = t, \quad \sigma_t = 1 - t$$

Then:

$$g(t) = \frac{(1-t)^2}{t^2} = \left(\frac{1}{t} - 1\right)^2$$

$$g^{-1}(y) = \frac{1}{1 + \sqrt{y}}$$

**Variance-preserving schedule.** Let us set:

$$\sigma_t = \sqrt{1 - t}, \quad \alpha_t = \sqrt{t}$$

Then:

$$g(t) = \frac{1-t}{t} = \frac{1}{t} - 1$$

$$g^{-1}(y) = \frac{1}{1 + y}$$

**Variance-exploding schedule.** Let us set:

$$\sigma_t = \sqrt{t}, \quad \alpha_t = 1$$

Then:

$$g(t) = t$$

$$g^{-1}(y) = \frac{1}{y}$$

## C EXTENDED RELATED WORK

**Other applications.** A wide range of reward guidance methods have been proposed (Chung et al., 2022; Abdolmaleki et al.; Ye et al., 2024; Yu et al., 2023; Bansal et al., 2023; He et al., 2023; Graikos et al., 2022), particularly focused on solving inverse problems such as Gaussian deblurring or inpainting. Many of the methods can be seen as various approximations of the posterior $p_{1|t}$. Here, we give a new way of sampling from $p_{1|t}$, potentially also opening new possibilities for these type of problems. However, note that the text-to-image setting we consider in this work comes with various challenges and constraints that are different than many of the settings these works consider: The reward models are neural networks themselves, i.e. we cannot query them out-of-distribution, and their gradients might not be informative. Further, the reward models are highly non-convex.

**Other approaches to reward alignment.** We briefly discuss other methods. The LATINO sampler (Spagnoletti et al., 2025) devises a scheme that iteratively noises a data point and then maps it back to a clean data point with a one-step sampler. The DEMON method (Yeh et al., 2024) uses stochastic SDE based sampling of diffusion models and considers the noise that is added as part of the SDE in a search space, i.e. they find the optimal noise to be added to an SDE. We could apply GLASS Flows also to this setting. Wu et al. (2024) introduce an auxiliary variable that is effectively a noisy version of a clean image and then sample both jointly via a Gibbs sampler. Krishnamoorthy et al. (2023) train a classifier-free guidance model where the conditioning variable $y$ is the reward or objective value. By setting that reward to be high, they then approximately sample from the distribution of high values.

**Other posterior approximations and related sampling methods.** Previous works have explored sampling from the posterior $p_{1|t}$, i.e. instead of just learning the mean via the denoiser actually sample from the posterior (De Bortoli et al., 2025; Elata et al., 2024; Chen et al., 2025a). For example, De Bortoli et al. (2025) explore learning the posterior posterior $p_{1|t}$ via a generative model trained via scoring rules allowing it to sample discrete-time transitions. In our experiments, we have also explored first sampling $z \sim p_{1|t}$ via GLASS Posterior Flows and then taking a conditional transition $p_{t'|t}(x_{t'}|x_t, z)$. However, we found it to lead to significantly worse performance than going "directly" to $x_{t'}$ by sampling a GLASS transition (note that our setting is different as we reparameterize an existing model instead of training a new one as in (De Bortoli et al., 2025; Shaul et al., 2025)). Similarly, Gaussian mixture flow matching (Chen et al., 2025a) approximates the posterior $p_{1|t}$ via a Gaussian mixture. This induces more stochasticity into a flow model and is shown to lead to performance improvements. Further, our method shares ideas from restart sampling Xu et al. (2023b), in that a variable is noised and then denoised via a flow again. However, our method is different in that the new noisy variable $\bar{x}_s$ still depends on $x_t$ up to $s = 1$. Further, our method does not really go back in time (i.e. it does not restart, the time $t^*$ where the neural network is queried can monotonically increase). Finally, our method is not approximate but theoretically exact.

**Scheduler changes for Gaussian probability paths.** Proposition 2 does not require the desired scheduler to match the pre-trained denoiser's scheduler. Scheduler changes have been studied previously, where reparameterization recovers the denoiser in this more limited setting of a single measurement (Lipman et al., 2024; Karras et al., 2022; Shaul et al., 2023; Pokle et al., 2023). GLASS Flows substantively extends this to multiple correlated Gaussian measurements.

**Detailed discussion of Transition Matching (TM) (Shaul et al., 2025).** TM is a general pre-training framework for *learning* inner flow matching models. Due to changes in the neural network architecture and the focus on pre-training, TM and GLASS Flows are different, complementary methods. However, the theoretically optimal transitions are indeed closely related as we explain here in more detail. Specifically, we discuss here the DTM supervision process (Shaul et al., 2025). We write it here via a continuous time variable $0 \le t \le 1$ using the convention of our work. In TM, the intermediate times are sampled via the CondOT probability path

$$X_t = t \cdot X_1 + (1 - t)X_0, \quad X_1 = z \sim p_{\text{data}}, X_0 \sim \mathcal{N}(0, I_d)$$

The training target is $Y = X_1 - X_0$, i.e. TM learns a flow matching model to sampling from the conditional distribution of $Y|X_t = x_t$. Note that for fixed $X_t = x_t$, it holds that

$$X_0 = \frac{x_t - tX_1}{1 - t} \quad \Rightarrow \quad Y = X_1 - X_0 = \frac{X_1 - x_t}{1 - t}$$

Hence, sampling $Y$ or sampling $X_1$ is equivalent - one can transform each variable into one another. Further, let us define the probability path for the TM model conditioned on $x_t$:

$$\begin{aligned}
\tilde{X}_s &= \tilde{\alpha}_s Y + \tilde{\sigma}_s \epsilon \quad \epsilon \sim \mathcal{N}(0, I_d) \\
&= \tilde{\alpha}_s \frac{X_1 - x_t}{1 - t} + \tilde{\sigma}_s \epsilon, \quad \epsilon \sim \mathcal{N}(0, I_d) \\
&= \underbrace{\frac{\tilde{\alpha}_s}{1 - t} X_1 + \tilde{\sigma}_s \epsilon}_{=: \bar{X}_s} - \frac{\tilde{\alpha}_s}{1 - t} x_t \\
&= \bar{X}_s - \frac{\tilde{\alpha}_s}{1 - t} x_t
\end{aligned}$$

where $\tilde{\alpha}_s, \tilde{\sigma}_s$ are schedulers for the inner TM model, i.e. Shaul et al. (2025) choose $\tilde{\alpha}_s = s$ and $\tilde{\sigma}_s = 1 - s$. The variable $\bar{X}_s$ is distributed according to the GLASS probability path in eq. (18) if we chose $t' = 1$ and $\bar{\alpha}_s = \tilde{\alpha}_s/(1-t)$ and $\bar{\sigma}_s = \tilde{\sigma}_s$:

$$p_s(\bar{x}_s|x_t, z) = \mathcal{N}(\bar{x}_s; \bar{\alpha}_s z, \bar{\sigma}_s^2 I_d)$$

where we used that $\bar{\gamma} = 0$ in this case. Therefore, if we simulate a GLASS trajectory with these parameters ($t' = 1, \bar{\sigma}_s = \tilde{\sigma}_s, \bar{\alpha}_s = \tilde{\alpha}_s/(1-t)$), we obtain that the transformed GLASS trajectory

$$\tilde{X}_s = \bar{X}_s - \frac{\tilde{\alpha}_s}{1-t}x_t$$

is a trajectory obtained from TM/DTM. This elucidates the connection between DTM TM and GLASS Flows.

**Detailed discussion of TADA (Chen et al., 2025b).** We provide an extended discussion of how the TADA method relates to this work. Specifically, we rewrite a simplified argument from (Chen et al., 2025b) in the notation of this work to showcase the connection. Specifically, we focus here on a simple case of $N = 2$, i.e. augmenting the data space $z \in \mathbb{R}^d$ with a single variable $p \in \mathbb{R}^d$, and where the terminal point $\tilde{\mathbf{x}}_1 = (z, p_1)$ in the state space is sampled independently, i.e. $z \sim p_{\text{data}}$ and $p \sim \mathcal{N}(0, I_d)$ (for the argument in full generality, we refer to (Chen et al., 2025b)). The probability path in this case is given by

$$\tilde{\mathbf{x}}_t = (\mu_t \otimes I_d)\tilde{\mathbf{x}}_1 + (L_t \otimes I_d)\epsilon, \quad \epsilon = (\epsilon_1, \epsilon_2)^T, \epsilon_1, \epsilon_2 \sim \mathcal{N}(0, I_d)$$

The core realization connecting it to our work is now that that $\tilde{\mathbf{x}}_t = (x_t, p_t)$ consists of two components $x_t$ and $p_t$ that are both in themselves noisy (correlated) Gaussian measurements of $z$. This follows simply from the fact $p_1$ is Gaussian and $\epsilon = (\epsilon_1, \epsilon_2)$ is Gaussian and linear transformations of Gaussian are again Gaussian. Hence, we are in a similar setting as in section 4.2.1. In fact, Chen et al. (2025b) applied a similar argument to recover the denoiser from a pre-trained flow matching or diffusion model as in section 4.2.1 (see (Chen et al., 2025b, Proposition 3.1)). This showcases the close connection of the mathematical principles enabling (Chen et al., 2025b) and this work.

# D    EXPERIMENTS

In this section, we provide details for experiments and present further experimental results.

## D.1    SAMPLING FROM THE POSTERIOR AND VALUE FUNCTION ESTIMATION (SECTION 6.1)

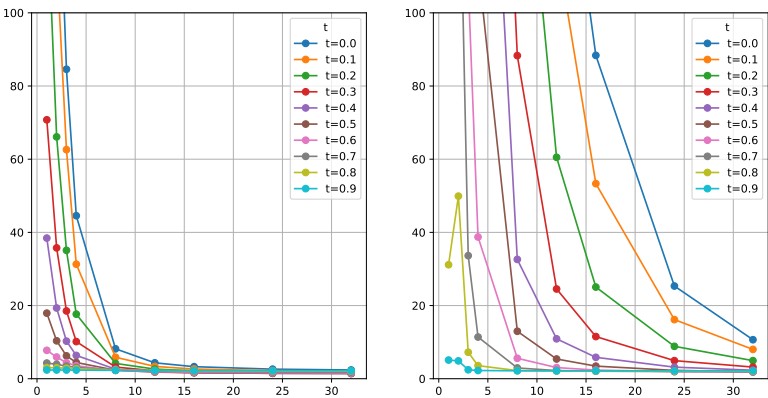

Figure 4: Detailed results for fig. 2 (Middle). Comparing the performance of sampling the posterior $p_{1|t}$ via GLASS Flows (Ours) and SDE (DDPM) sampling. Ablate over different times $t$ and sampling steps. GLASS Flows achieve significantly lower FID for lower number of sampling steps than DDPM sampling.

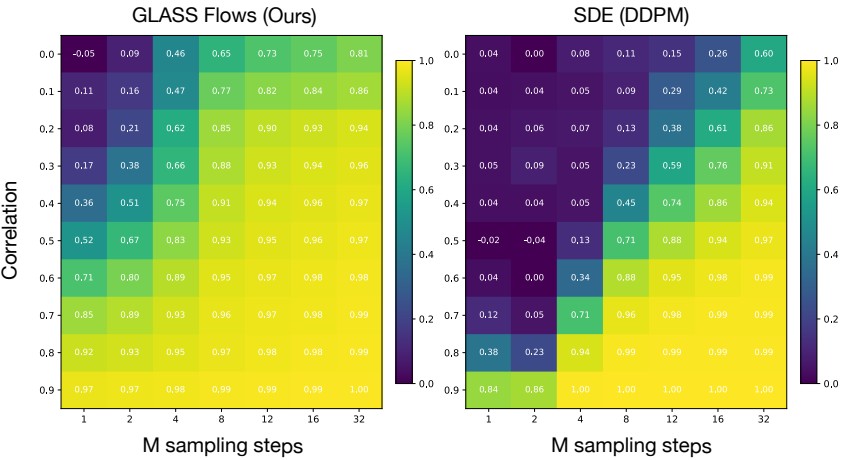

Figure 5: Detailed results for fig. 2 (Right). Comparing the performance of estimating the value function $V_t(x)$ via sampling the posterior $p_{1|t}$ via GLASS Flows (Ours) and SDE (DDPM) sampling via correlation. Experiment performed for different times $t$ and sampling steps $M$. GLASS Flows achieve significantly higher correlation for lower number of steps than DDPM sampling. Ground truth is measured via 200 samples with 200 simulation steps of ODE/SDE.

## D.2 SAMPLING RESULTS

Table 3: GenEval results

| Algorithm | Overall | 1 object | 2 objects | colors | color attr. | position | counting |
|---|---|---|---|---|---|---|---|
| ODE Sampling | 0.6327 | 0.9843 | 0.8005 | **0.7154** | 0.4313 | **0.1900** | 0.6719 |
| SDE (DDPM) | 0.4435 | 0.6938 | 0.4596 | 0.5186 | 0.2720 | 0.1200 | 0.5969 |
| GLASS (DDPM) | **0.6357** | 0.9812 | **0.8030** | 0.7074 | **0.4746** | 0.1475 | 0.7031 |
| GLASS ($\rho = 0.4$) | 0.6304 | **0.9844** | 0.7904 | 0.7101 | 0.4449 | 0.1400 | **0.7125** |

Table 4: Sampling evaluation for SiT and FLux models using various sampling algorithms introduced in this work. We use 50 total neural network evaluations for all experiments and 5 transitions (i.e. 10 simulation steps for each transition).

| Algorithm | SiT | Flux | | | |
|---|---|---|---|---|---|
| | FID | CLIP | Pick | HPSv2 | IR |
| ODE Sampling | **2.34** | 33.82 | **22.80** | 0.291 | 1.060 |
| SDE (DDPM) | 4.36 | 33.70 | 22.55 | 0.287 | 1.017 |
| GLASS (DDPM) | 2.58 | 33.81 | 22.73 | 0.273 | **1.079** |
| GLASS ($\rho = 0.4$) | 2.54 | **33.90** | 22.72 | **0.293** | 1.049 |

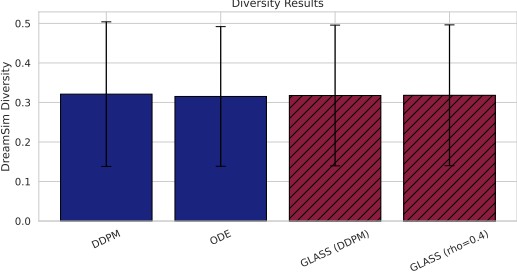

Figure 6: Evaluating diversity of samples from various sampling scheme. All 3 sampling approaches (DDPM, ODE, GLASS) show very similar results for diversity. This is consistent with theory as all 3 approaches should sample from the same distribution. We evaluate the model with 100 total NFEs on GenEval prompts (we take more NFEs than in fig. 3 to reduce discretization error of the SDE). We take 8 samples per prompt and measure the average DreamSim (Fu et al., 2023) similarity between two samples (error bars equal average standard deviation of samples of prompt).

## D.3 ABLATION OVER CORRELATION PARAMETER $\rho$

We perform further experiments in this section ablating the correlation parameter $\rho$. We choose two strategies for ablation: First, a constant correlation parameter $\rho$ is chosen across all transitions from $t \to t'$ independent of $t, t'$. Second, we choose $\rho$ as time-varying based on the DDPM schedule (see proposition 1):

$$\rho = \left( \frac{\alpha_t \sigma_{t'}}{\sigma_t \alpha_{t'}} \right)^\kappa$$

and we ablate over the parameter $\kappa \geq 0$. We use the FLUX model and measure GenEval performance for prompt adherence/generation quality and DreamSim diversity (Fu et al., 2023) as a measure of diversity. We present results in fig. 7. The most striking results is that **GLASS Flows achieves ODE-level performance for almost all correlation schedules and differences of performance are relatively minor.** A constant correlation schedule of $\rho = 0.4$ performs best in GenEval performance and has relatively high sample diversity. Therefore, we choose $\rho = 0.4$ in subsequent

experiments. For the ablation over $\kappa$, DDPM ($\kappa = 1$) performs very high (just diversity is slightly higher for $\kappa = 2$). Due to the wide use of DDPM and its theoretical importance as a time-reversal, we also use DDPM in subsequent experiments.

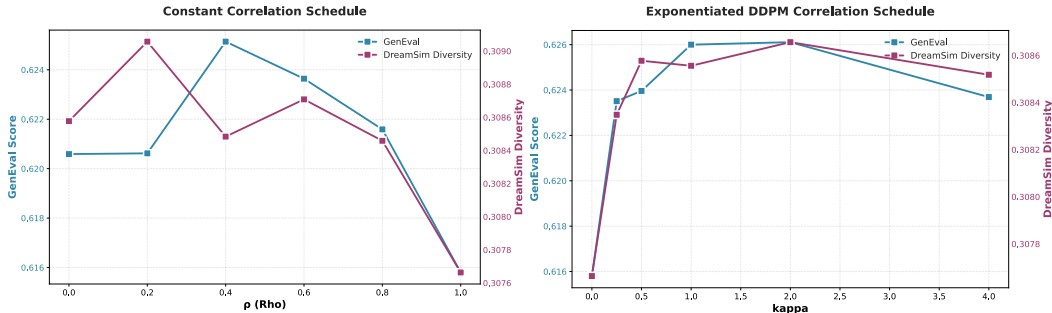

Figure 7: Ablation experiment of correlation schedule $\rho$. Left: Constant correlation $\rho$ across all transitions. Right: Time-dependent correlation schedule given by $\rho = \left( \frac{\alpha_t \sigma_{t'}}{\sigma_t \alpha_{t'}} \right)^{\kappa}$ - note that $\kappa = 1$ corresponds to the DDPM schedule (see proposition 1).

### D.4 REWARD GUIDANCE

In this section, we explore using reward guidance with GLASS Flows to improve text-to-image alignment and also discuss the challenges in assessing reward guidance in the context of text-to-image generation. We note that reward guidance has so far been mainly used for inverse problems (Chung et al., 2022; He et al., 2023) and only few works explore it to improve text-to-image alignment for large-scale models (Singhal et al., 2025; Eyring et al., 2024), the application we focus on in this work. As an intermediate reward, we use the common model of $r_t(x) = r(\text{VAE}(D_t(x)))$ where VAE is decoder of latent diffusion model. Computing $\nabla r_t(x)$ via backpropagation led to out-of-memory errors (mixed precision on A100 80GB memory GPUs). To remedy this, we make two modifications: First, we use lower resolution images (size $676 \times 676$ instead of size $768 \times 1360$) to remove a memory bottleneck at the output the VAE. Second, we detach the denoiser $D_t(x)$ from the computation graph and compute the gradient at that point. This is a common technique in the context of linear inverse problems (He et al., 2023). This allows us to compute a gradient at every step with reasonable computational overhead (as the velocity field $u_t$ model is significantly bigger than the VAE or the reward model). Further, we then set the guidance strength $c_t$ in eq. (10) as $c_t = \lambda \cdot \nu_t^2/2$ for a hyperparameter $\lambda$. Further, we set $c_t = 0$ for $0.2 \leq t \leq 0.7$ for numerical stability, i.e. we only apply guidance in the interval $[0.2, 0.7]$. In table 5, we present the results for $\lambda = 0.4$. However, the guidance strength $\lambda$ has varying effects on different methods. Therefore, we vary the guidance strength with ImageReward and plot effects on GenEval performance in fig. 8. As one can see, all methods can increase the ImageReward value arbitrarily with artifacts appearing for high guidance strength. GLASS Flows achieves the highest performance GenEval for the same ImageReward values.

### D.5 FURTHER RESULTS FOR FEYNMAN-KAC-STEERING

In fig. 9, we plot results for Feynman-Kac Steering with GLASS Flows on the PartiPrompts benchmark. This further confirms the results from section 6.3 that GLASS Flows improve the state-of-the-art performance.

Table 5: Reward guidance results on GenEval prompts. $N = 50$ simulation steps. The best value in each column is **bolded**, and the second best is underlined. Reward guidance with GLASS Flows improves both GenEval score and the reward of interest, while flow guidance leads to decreased performance on GenEval.

| Algorithm | CLIP | | Pick | | HPSv2 | | IR | |
|---|---|---|---|---|---|---|---|---|
| | CLIP | GenEv. | Pick | GenEv. | HPSv2 | GenEv. | IR | GenEv. |
| Flow baseline | 34.9 | **63.8** | 23.4 | **63.8** | 0.302 | 63.8 | 0.884 | 63.8 |
| SDE baseline | 34.7 | 57.0 | 22.9 | 57.0 | 0.280 | 57.0 | 0.621 | 57.0 |
| Flow guidance | **37.3** | 63.0 | **24.3** | 63.4 | **0.320** | 63.0 | **1.387** | 62.7 |
| SDE guidance | 36.9 | 60.1 | 23.5 | 61.2 | 0.294 | 60.9 | 1.267 | 61.3 |
| GLASS guidance | 36.6 | 63.4 | 23.9 | 63.6 | 0.314 | **63.9** | 1.315 | **64.7** |

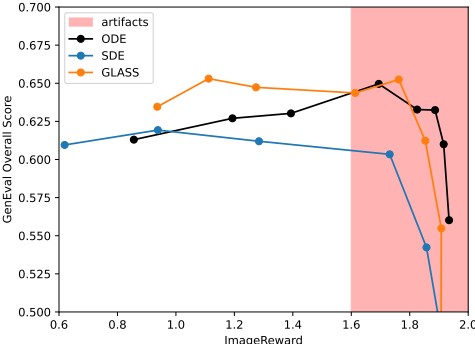

Figure 8: Varying reward guidance strength across different methods on GenEval benchmark with reward ImageReward. By increasing the guidance strength, we can increase ImageReward. GLASS Flows has higher performance on GenEval performance for the same ImageReward value. High guidance strengths lead to image artifacts that are not properly captured by our metrics.

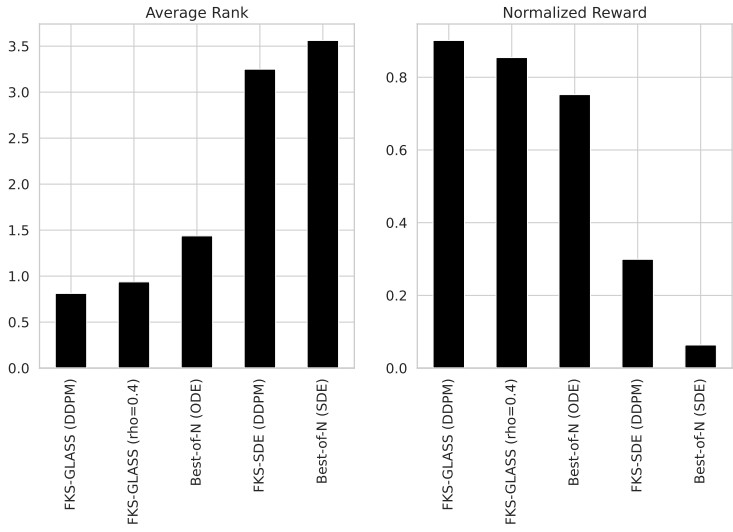

Figure 9: Inference-time reward alignment results on PartiPrompts benchmark. For each reward model (Clip, Pick, HPSv2, ImageReward), we run reward alignment with difference methods and evaluate across all reward models (i.e. this gives us $16 = 4 \times 4$ values). Left: We take the 16 values, rank the methods, and take the average rank. Right: We take the average normalized reward value (normalized via min and max observed).

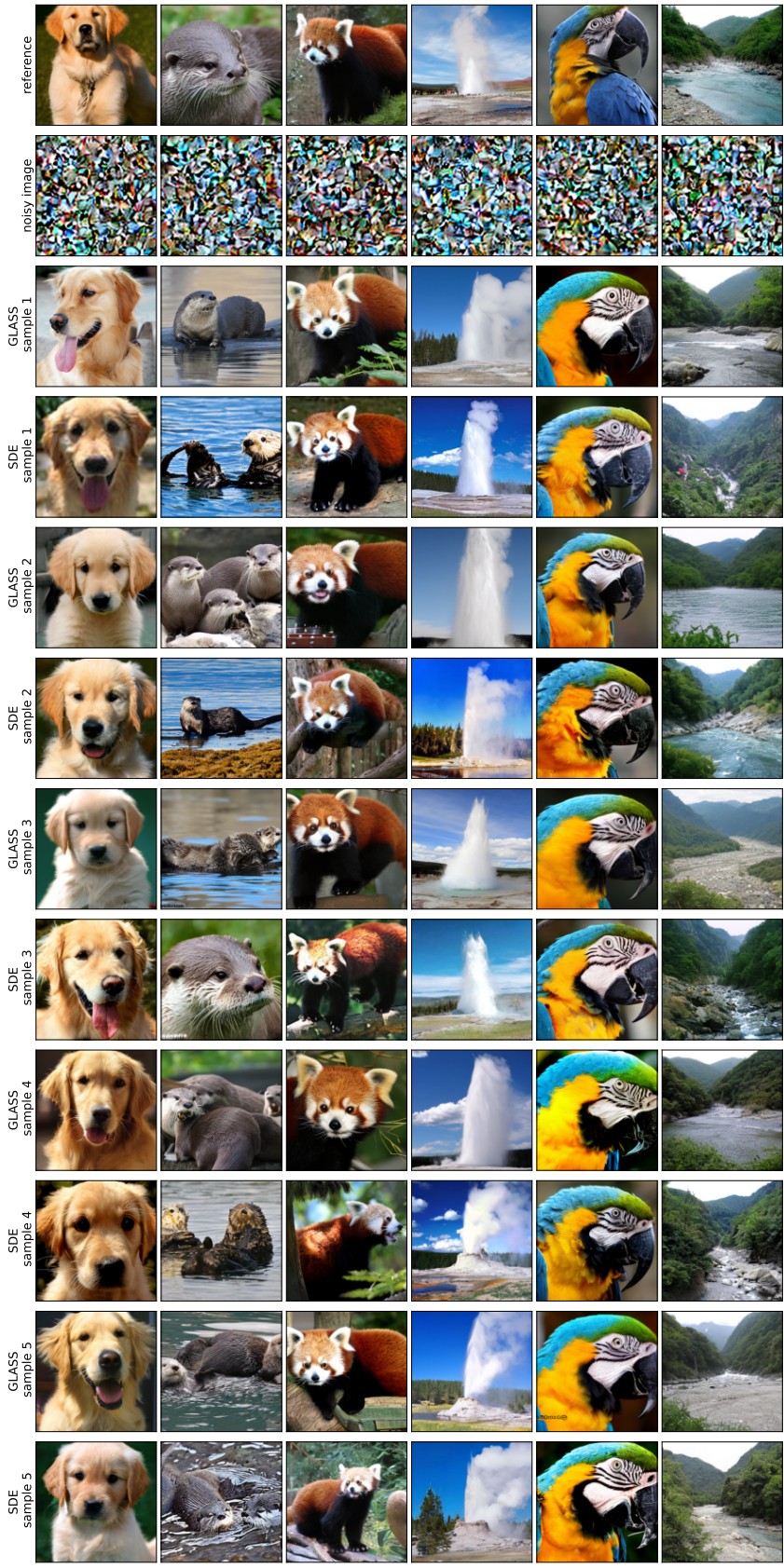

Figure 10: Various samples from the posterior $p_{1|t}$ via GLASS Flows using $M = 200$ simulation steps. Both GLASS Flows and the SDE sample from the posterior given the noisy image.

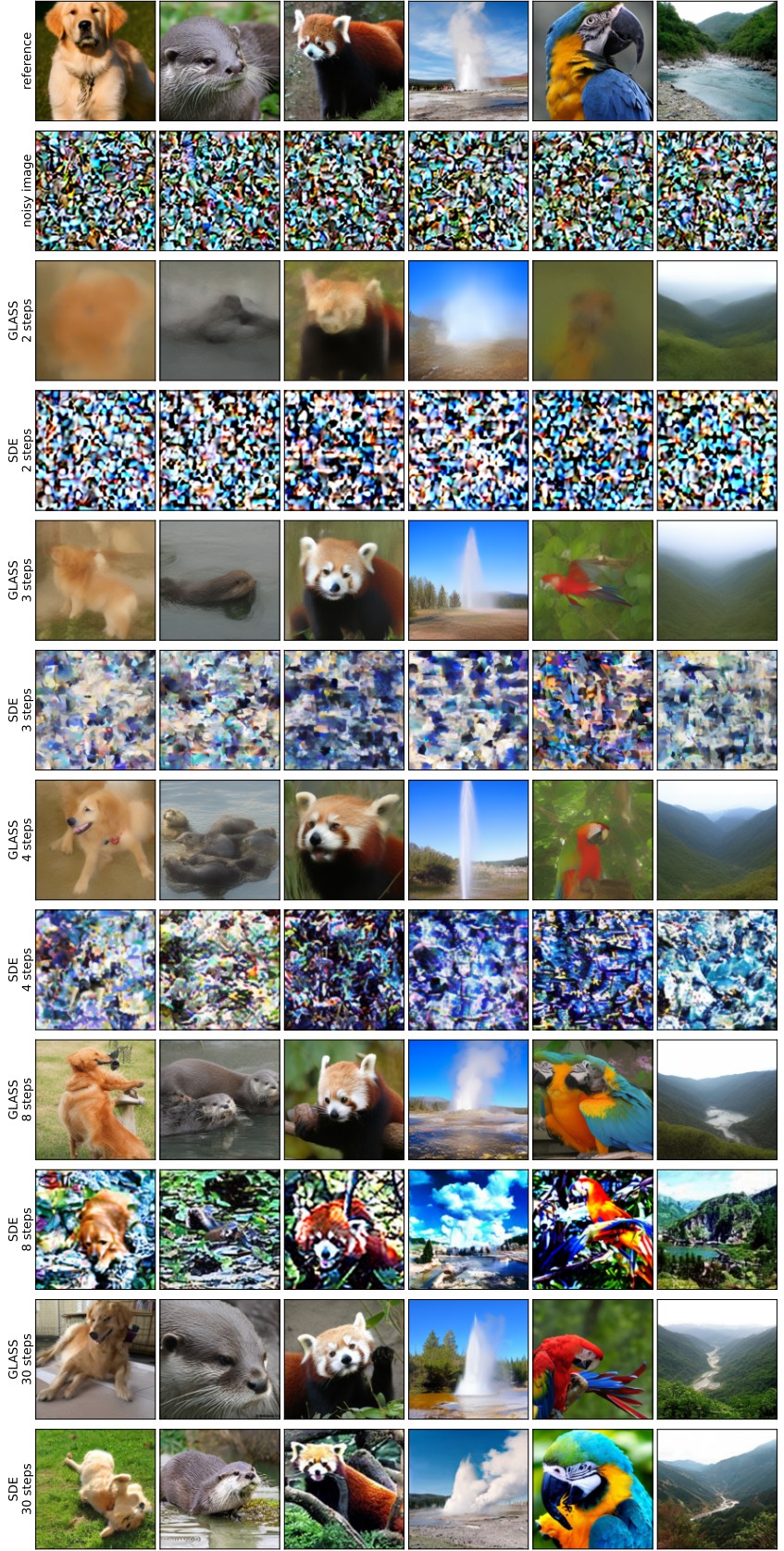

Figure 11: Posterior recovery for $t = 0.05$ for various number of simulation steps $M$. As one can see, GLASS Flows achieve significantly better performance for low $M$ than the SDE/DDPM sampling.

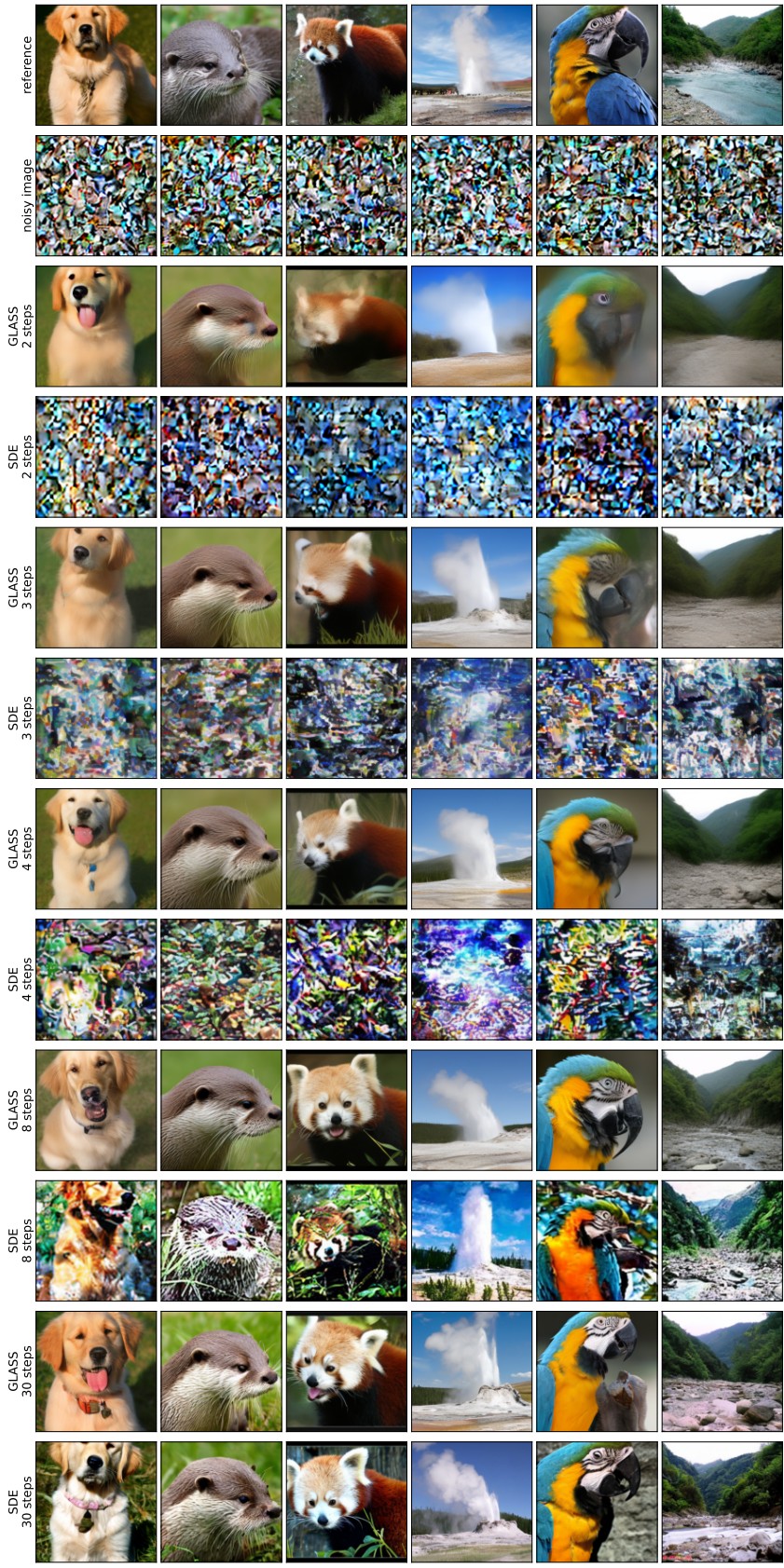

Figure 12: Posterior recovery for $t = 0.15$ for various number of simulation steps $M$. As one can see, GLASS Flows achieve significantly better performance for low $M$ than the SDE/DDPM sampling.

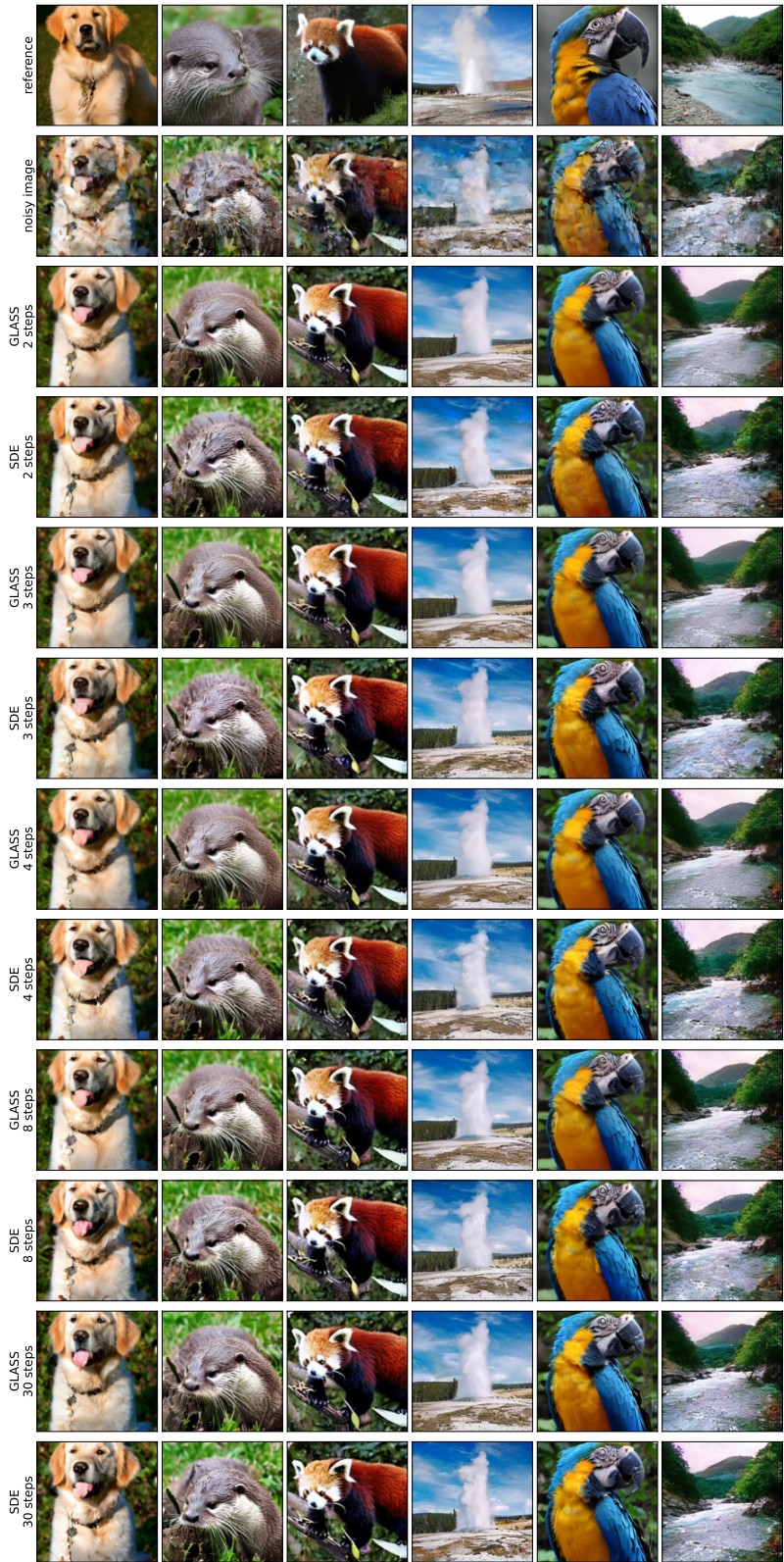

Figure 13: Posterior recovery for $t = 0.7$ for various number of simulation steps $M$. As $t$ is close to 1, the uncertainty/variance of $p_{1|t}$ is very low. Hence, also with low number of simulation steps, a reasonable performance is achieved regardless of the method (best compared with fig. 11, fig. 12).

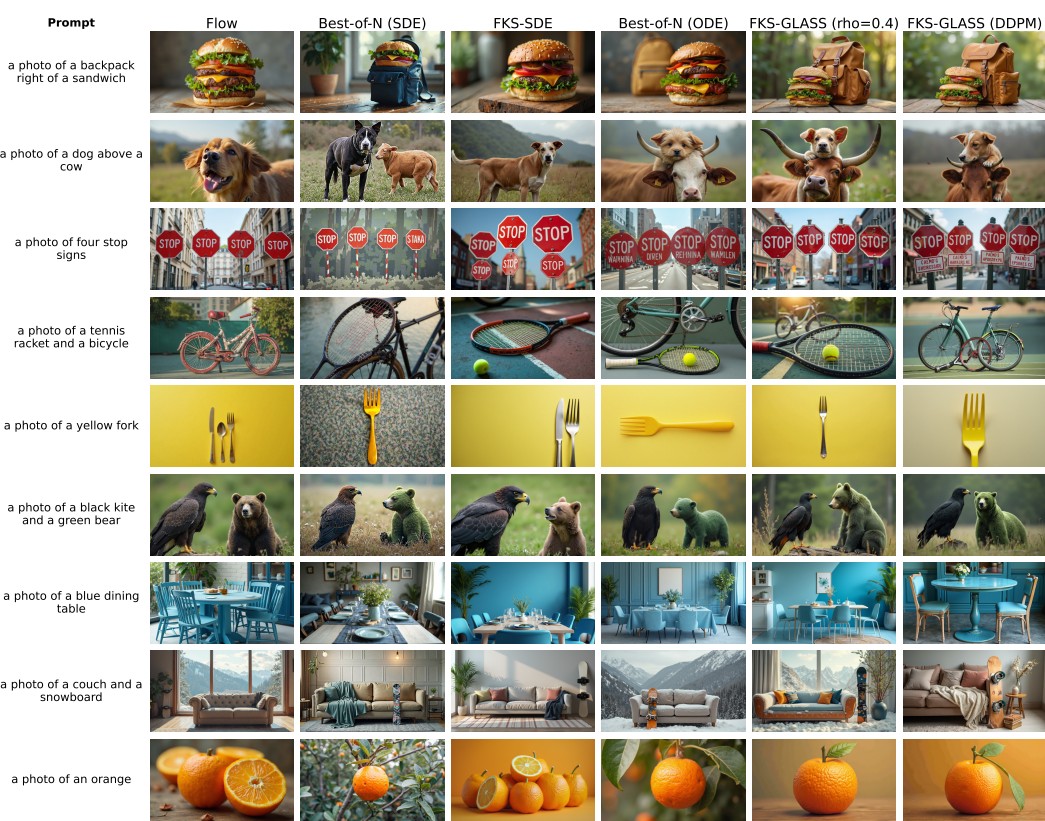

Figure 14: Examples for reward alignment Sequential Monte Carlo experiment on GenEval (see section 6.3).

