# OpenReview forum: "GLASS Flows: Efficient Inference for Reward Alignment of Flow and Diffusion Models"
_ICLR.cc/2026/Conference — ICLR 2026 Oral_

### Official Review · Reviewer_1Ktu · 2025-10-31

**Soundness:** 4
**Presentation:** 3
**Contribution:** 4
**Rating:** 8
**Confidence:** 2

**Summary:**

This paper proposes **GLASS Flows**—a sampling paradigm that lets a pre‑trained flow/diffusion model simulate **stochastic Markov transitions** using only **ODE integration**. The key idea is to view a transition $x_t \rightarrow x_{t'}$ as its own conditional generation problem and to construct an “inner” flow-matching model $u_s(\bar x_s\mid x_t,t)$ whose terminal state $\bar x_{s=1}$ is distributed as the desired transition $p_{t'|t}(\cdot\mid x_t)$. Technically, the paper defines a **GLASS transition** by a bivariate Gaussian joint
$(X_t,X_{t'})\sim\mathcal N(z\mu,\Sigma)$ with a tunable correlation $\rho$ across time (no cross‑dimension correlation), inducing $p_{t'|t}$. In practice, GLASS is a drop‑in replacement wherever reward‑time algorithms need stochastic transitions: (i) posterior sampling and value‑function estimation $V_t$, (ii) Sequential Monte Carlo (SMC; Feynman–Kac steering), and (iii) guidance. Experiments on SiT/DiT ImageNet‑256 and FLUX (768×1360) show better few‑step posterior sampling (lower FID at small inner steps $M$), higher correlation in $V_t$ estimation, and SMC reward gains without sacrificing GenEval alignment; with $M=1$ the method reduces to DDIM.

**Strengths:**

- The paper empirically demonstrates that GLASS transitions retain ODE‑level efficiency yet produce stochastic branches needed for SMC/search. This answers a real bottleneck noted in prior work: practitioners prefer ODE/probability‑flow sampling for speed (e.g., EDM design space) while SMC/search need sampling from $p_{t'|t}$ (typically SDE‑based).
- Casting a transition $p_{t'|t}$ as an inner flow and showing that its denoiser is just the sufficient statistic re‑timing of the existing denoiser is elegant and broadly applicable. It explains in one stroke how to obtain stochasticity from an ODE‑driven model without re‑training. The mapping via $S(x)=\frac{\mu^\top\Sigma^{-1}x}{\mu^\top\Sigma^{-1}\mu}$ and $t^\*$ is simple to implement and leverages any pre‑trained flow/diffusion network.
- Plug‑and‑play application to Feynman–Kac SMC (GLASS‑FKS) and reward guidance improves reward metrics while preserving GenEval alignment, and does so on strong modern backbones (SiT/DiT and FLUX). This makes the method attractive to practitioners who already deploy ODE sampling but need stochastic branches.{index=6}

**Weaknesses:**

The statement that GLASS “eliminates the trade-off between efficiency and stochasticity” is strong, yet Tables 1 and 4 show roughly comparable rather than strictly superior performance. Missing wall-clock time and variance analyses weaken this claim.

The influence of correlation $ \rho $ and inner-scheduler choices (CondOT vs. alternatives) is underexplored. Additional ablations could clarify their role in balancing stochasticity and sample quality.

**Questions:**

1. How does GLASS handle near-singular covariance matrices when $ |\rho| \approx 1 $? Are any clipping or jitter techniques applied to maintain numerical stability?

2. Is $ g(t) = \sigma_t^2/\alpha_t^2 $ guaranteed to be invertible under learned or non-monotonic schedules? If not, how is $ t^\* $ computed?

3. Could $ \rho $ be learned or dynamically adjusted during inference to balance exploration and stability, especially in SMC or value-guided sampling?

---

> ### Author Response · Authors · 2025-11-21
>
> We sincerely thank the reviewer for insightful comments including the constructive feedback allowing us to improve our work. We made additional experiments to address the concerns of the reviewer and outline responses below.
>
> **Influence of correlation parameter $\rho$.** We added **several ablation experiments** over various schedules for correlation parameter $\rho$ (see new section in appendix D.3.). There, we addressed the choice of the correlation parameter (see appendix D.3). We evaluated generation quality/prompt adherence and diversity, as suggested by the reviewer. GLASS Flows achieves ODE-level performance for almost all correlation schedules and differences of performance are relatively minor. The parameter $\rho=0.4$ was the parameter achieving highest performance. Hence, we chose this parameter in subsequent experiments. We refer to appendix D.3. for details.
>
> **Evaluation.**
> >Tables 1 and 4 show roughly comparable rather than strictly superior performance. Missing wall-clock time and variance analyses weaken this claim.
>
> We consider our results to be significant for several reasons:
> - **Extensive experiments:** Our experiments were performed on 2 benchmarks (GenEval + PartiPrompts) and each benchmark was tested for 4 different reward models. In all of the experiments, GLASS outperforms other methods by a meaningful margin. We note that many previous works only use 1 reward model and one set of prompts.
> - **No tuning:** We use the same parameters across all these settings, i.e. we did not tune the parameters of GLASS Flows to overfit to a single setting but rather use the same setting across all benchmarks and reward models.
> - **Real-world task with SOTA models:** We use state-of-the-art models on real tasks and high resolution images (700x1300 resolution). This makes experiments significantly more compute- and time-intensive.
>
> We agree that wall-clock time and variance across results are indeed important aspects. We addressed this in the following way:
> - Wall-clock time: All experiments we perform are matched in the number of function evaluations of the diffusion model. The reparameterizations we perform in Algorithm 1 are negligible (we highlighted that now again in the main text). Hence, sampling with GLASS Flows or with ODE sampling has the same wall clock time. For table 1 (reward steering experiments), the only difference in wall clock time comes from the difference in Feynman-Kac Steering (FKS) or Best-of-N. This is unrelated to GLASS Flows: FKS queries intermediate rewards, which Best-of-N does not. However, the diffusion model is the largest neural network by a significant margin compared to the reward models (e.g. FLUX has ~100x more parameters than CLIP). Therefore, this overhead is negligible. We hope this explains why all our experiments are compute-matched and have the same wall clock time up to negligible margin.
> - Variance analyses: We agree that a variance analysis would be instructive. However, the reward steering experiments are very compute heavy: We need to perform N=8 samples per prompt on large-scale models for high-resolution images on large sets of prompts. Instead of running the same benchmark for the same reward model with different seeds, we focused on running experiments on 2 benchmarks with different reward models. As GLASS Flows outperforms all other methods in all settings, we consider this a clear signal that GLASS is indeed a significantly better method.
>
> **Questions.**
> > Are any clipping or jitter techniques applied to maintain numerical stability?
>
> We added a new section (appendix B.3.) on how to ensure numerical stability. We describe several steps to account for edge cases and make everything numerically stable. We also updated Algorithm 1 to account for the edge case $s=0$.
>
> >Is g(t) guaranteed to be invertible under learned or non-monotonic schedules? If not, how is computed?
>
> In our work, we assume that the schedulers $\alpha_t$ (resp. $\sigma_t$) are monotonically increasing (resp. decreasing), see section 2. As suggested by the reviewer, for non-monotonic schedulers, the function $g(t)$ would not be invertible. However, for any valid diffusion model, the function g would still be surjective from $[0,1]$ to $[0, \inf)$. So one could then simply pick one possible $t*$  such that $g(t^*)$ has the desired value. So monotonicity is more an assumption for mathematical convenience but not strictly necessary. Further, all diffusion models have an invertible signal-to-noise ratio (SNR) and therefore this assumption is reasonable.
>
> >Could rho be learned or dynamically adjusted during inference to balance exploration and stability, especially in SMC or value-guided sampling?
>
> Yes, it could be. We made rho a constant in time for simplicity in the notation but we can choose rho dynamically and time-dependent as a function of $t,t’$. This is a great point and we highlight this now as future work.
>
> Thank you again for your helpful feedback and response.

---

> ### Comment · Reviewer_1Ktu · 2025-11-24
>
> Thanks to the author for the clarification and the updated algorithm; my confusion and concerns have been resolved. I believe this is an excellent paper and deserves to be accepted.

---

### Official Review · Reviewer_56nT · 2025-11-01

**Soundness:** 3
**Presentation:** 3
**Contribution:** 3
**Rating:** 6
**Confidence:** 3

**Summary:**

The paper proposes **GLASS Flows**, a plug-and-play sampler for diffusion/flow-matching models that realizes stochastic transitions via an inner ODE (instead of an SDE). By reparameterizing a pretrained denoiser into a ``GLASS denoiser" using sufficient statistics, the method constructs a transition kernel \(p_{t'|t}\) without retraining, unifying ODE-level quality with SDE-like randomness. This improves posterior sampling and downstream inference-time reward adaptation (e.g., SMC/FKS, reward guidance, search). At matched NFE, GLASS approaches ODE quality, outperforms SDE, and yields stronger prompt following/reward alignment on large T2I models.

**Strengths:**

* Theory to practice bridge: Closed-form construction of a stochastic transition via an inner ODE.
* Drop-in practicality: No finetuning; works with CFG, SMC/FKS, reward guidance, and value-estimation pipelines.
* Quality & alignment: Consistent gains on prompt-following and reward-driven metrics at the same NFE, while retaining randomness for search/SMC.
* Clear complexity lens: Cost summarized by NFE = \(K \times M\) (transitions × inner steps), enabling fair compute-matched comparisons.

**Weaknesses:**

1. **Choice of $\rho$.**
The paper fixes $\rho = 0.4$  for most experiments, but this parameter directly controls the stochasticity of the transition. It would be helpful to show how varying $\rho$ in $[0, 1]$ affects generation quality, prompt adherence, and diversity, and to explain why $\rho = 0.4$  was chosen. In addition, please comment on whether different models (e.g., SiT vs. FLUX) show similar sensitivity to $\rho$ or require different optimal values.

2. **Under-evaluated relation to DTM/Transition-Matching.**
DTM is claimed as the  $\rho = 1$ special case, but there may lacks head-to-head comparison.

3. **Positioning vs post-training with reward models (RL/DPO/GRPO).**
Recent post-training approaches such as GRPO[1,2] or DPO also leverage reward models for alignment, but differ by updating model parameters rather than modifying inference. It would be helpful if the authors could clarify the conceptual connection and distinction between these paradigms, and possibly comment on whether GLASS could be complementary to such training-based methods or directly compared under similar reward setups.

[1] DanceGRPO: Unleashing GRPO on Visual Generation

[2] MixGRPO: Unlocking Flow-based GRPO Efficiency with Mixed ODE-SDE

**Questions:**

Please see the weaknesses.

---

> ### Author Response · Authors · 2025-11-21
>
> We sincerely thank the reviewer for insightful comments including the constructive feedback allowing us to improve our work. We made additional experiments to address the concerns of the reviewer and outline responses below.
>
> **Choice of rho.**
>
> >It would be helpful to show how varying rho affects generation quality, prompt adherence, and diversity, and to explain why rho=0.4 was chosen.
>
> We added a new section to the work addressing the choice of the correlation parameter $\rho$ and we performed **additional ablation experiments in appendix D.3** over various schedules for the correlation parameter $\rho$. We evaluated generation quality/prompt adherence and diversity, as suggested by the reviewer. GLASS Flows achieves ODE-level performance for almost all correlation schedules and differences of performance are relatively minor. The schedule of constant correlation parameter rho=0.4 was the parameter achieving highest performance. We hope this clarifies why we chose this parameter in subsequent experiments.
>
> >Please comment on whether different models (e.g., SiT vs. FLUX) show similar sensitivity to or require different optimal values.
>
> Indeed, the choice of the correlation parameter rho is dependent on the setting. We highlight this now in section 6.2.
>
> **Relation to DTM/Transition-Matching.**
> >DTM is claimed as the special case, but there may lacks head-to-head comparison.
>
> Thank you for raising this point. It is true that there is a theoretical connection between TM and GLASS Flows. We added a **detailed discussion in appendix C explaining the relationship**. However, TM is not a special GLASS Flow due to important practical differences. In particular, we did not compare our work to Transition Matching (TM) for several reasons:
> - Different goals: Our work focuses on inference-time modifications of current SOTA pre-trained FM models. Transition Matching is a different pre-training paradigm.
> - No available SOTA models for TM: SOTA Transition Matching models like FLUX trained on web-scale data are not available. The largest available pre-trained TM models are trained on Shutterstock data (not web-scale data) and performance is significantly worse for these TM models compared to FLUX. Our work aims to improve inference-time scaling of current state-of-the-art models.
> - TM only samples approximations of the GLASS transition: TM significantly modifies the neural network architecture to learn the transitions approximately - the transitions are performed independently per patch of an image. Hence, TM does not actually sample the GLASS transitions that we study in this work but rather approximations of it.
>
> We hope this explains that while there is a theoretical connection, a comparison would not be possible before SOTA TM models are available.
>
>
> **Positioning vs post-training with reward models (RL/DPO/GRPO).** Thank you for bringing up reward fine-tuning methods that are indeed closely related. To address this, we **added a paragraph in section 5 to discuss the connection** of reward fine-tuning and GLASS Flows. We briefly summarize it here. GLASS has 2 different important synergies with these models:
> 1. **GLASS could be used to accelerate reward fine-tuning:** Many reward fine-tuning algorithms - including the ones cited by the reviewer - require the use of DDPM/SDE sampling for exploration during training [1,2,3,4]. However, this is very inefficient - as we show in our GLASS Flows work. Therefore, one can potentially accelerate reward fine-tuning methods via GLASS Flows by replacing the slow SDE sampling with GLASS Flows. This is a completely new application of GLASS Flows that could be explored in the future. Partial replacement of SDEs with ODEs was already explored in [2] and we could extend this idea significantly with GLASS Flows.
> 2. **GLASS Flows can be applied to fine-tuned models:** One can also apply inference-time scaling with GLASS Flows to fine-tuned models. Many of these models learn a FM model of reward-tilted distribution. Hence, GLASS Flows is equally valid to be applied to reward fine-tuned models and can be applied to improve reward fine-tuned models.
>
> Further, we note that a direct comparison is hard to perform in a fair way, however, as the amount of compute used for fine-tuning has a very high fix cost (but inference time is the same), while inference-time scaling increases the cost per sample but has no additional cost training cost. We hope that the above clarifies that both approaches complement each other and there are natural synergies. We highlight this in our work now with an additional paragraph (see section 5).
>
> Thank you again for your helpful feedback and response.
>
> [1] DanceGRPO: Unleashing GRPO on Visual Generation
>
> [2] MixGRPO: Unlocking Flow-based GRPO Efficiency with Mixed ODE-SDE
>
> [3] FlowGRPO: Flow-GRPO: Training Flow Matching Models via Online RL
>
> [4] Adjoint Matching: Fine-tuning Flow and Diffusion Generative Models with Memoryless Stochastic Optimal Control

---

### Official Review · Reviewer_JMLt · 2025-11-01

**Soundness:** 4
**Presentation:** 3
**Contribution:** 4
**Rating:** 8
**Confidence:** 4

**Summary:**

This paper introduces GLASS Flows, a new inference-time sampling paradigm that constructs an inner flow matching model within a pre-trained one to efficiently simulate stochastic evolution, combining the stochasticity of SDEs with the efficiency of ODEs without any retraining, thereby significantly improving the performance and efficiency of flow and diffusion models.

**Strengths:**

The paper introduces an elegant sampling method (GLASS flow) that successfully enhances sample diversity without compromising quality.

This represents a fundamental improvement and holds significant potential for integration into a wide range of reward alignment frameworks.

**Weaknesses:**

The paper lacks a direct quantitative comparison of sample diversity between the proposed GLASS flow and standard SDE sampling. Providing metrics to substantiate the diversity claims would strengthen the paper significantly.

**Questions:**

1.  **On SDE Performance in Few-Step Sampling (Figure 2):**
    Could the authors elaborate on the poor performance of SDE sampling in the few-step regime, as shown in Figure 2? Typically, reducing sampling steps in diffusion models results in blurriness or structural artifacts. However, the samples shown (e.g., in Figure 9, which seems related) exhibit high-frequency residual noise, suggesting the denoising process is incomplete rather than just imprecise. This seems counter-intuitive. [1] also analyzed the poor image quality produced by time-reversal SDE sampling and proposed an improvement that enables SDE sampling to generate high-quality images.


2.  **Baselines for Few-Step Sampling:**
    * For the experimental setup in Figure 2 (middle), what is the FID-vs-steps curve for standard ODE sampling? This baseline is crucial for contextualizing the performance of GLASS.
    * How does GLASS compare against other state-of-the-art few-step solvers, such as DPM-Solver++ or UniPC? These methods are specifically designed for high-fidelity generation in the low-NFE regime, and a comparison of FID scores would be highly informative.

3.  **Best-of-N Sampling (Table 1):**
    The results for "Best-of-N" sampling using GLASS are not presented in Table 1. Given that Best-of-N is a common technique in reward alignment, could the authors provide the performance metrics for a Best-of-N(GLASS) approach?

4.  **Applicability to Online RL Methods:**
    Could the proposed GLASS flow be integrated into online reinforcement learning algorithms for flow matching models, such as DDPO[2] or Flow-GRPO[3]? A discussion on its potential applicability, or any inherent limitations for these types of methods, would add significant value to the paper.

[1] Coefficients-Preserving Sampling for Reinforcement Learning with Flow Matching

[2] Training Diffusion Models with Reinforcement Learning

[3] Flow-GRPO: Training Flow Matching Models via Online RL

---

> ### Comment · Reviewer_JMLt · 2025-11-20
>
> I have one more question: in GLASS Flow, the correlation coefficient p determines the specific form of each sub-flow, and the randomness comes from the initial noise of each sub-flow. However, I did not see much discussion about how the initial noise is chosen. After fixing p, how much variation do different initial noise samples produce in the final generated images? And how does the diversity change as K varies?

---

> > ### Author Response · Authors · 2025-11-21
> >
> > We sincerely thank the reviewer for insightful comments including the constructive feedback allowing us to improve our work.
> >
> > **Sample diversity.**
> > >Providing metrics to substantiate the diversity claims would strengthen the paper significantly.
> >
> > Thank you for bringing up the point of diversity. We note that our paper did not make any claims about improved sample diversity in the original draft (more about stochastic evolutions).  We agree though that diversity is important and we performed additional experiments to assess diversity. Specifically, for each prompt, we sample $N$ times from each method and take the DreamSim diversity. In appendix D.2, we present the results and describe details for the experiments. As one can see, the diversity is the same for all 3 sampling methods (ODE/SDE/GLASS). This is consistent with the theory as SDE sampling, ODE sampling, and GLASS Flows all should sample from the same data distribution. Further, we ablated over the correlation parameter rho for GLASS Flows and assess the diversity in appendix D.3. The parameters we chose for rho in experiments (constant $\rho=0.4$) led to the best results.
> >
> > Further, there is also the diversity of rollouts from a fixed point x_t at time t in the flow/diffusion trajectory instead, i.e. what we commonly call stochastic evolution in our work. We show qualitatively the diversity (see figure 10) and also showed that observables of the posterior can be estimated better/faster with GLASS Flows (see section 6.1) - indicating that GLASS Flows can recover the exact posterior. Similarly, here, we do not claim that GLASS Flows is “more” diverse than DDPM sampling. In fact, they should sample from the same distribution in theory, as we prove. Rather, GLASS is a different - and much more efficient - approximation of the same transition. We hope that this clarification combined with our additional experiments address your concern about diversity.
> >
> > **Applicability to Online RL Methods.**
> > >Could the proposed GLASS flow be integrated into online reinforcement learning algorithms for flow matching models, such as DDPO or Flow-GRPO?
> >
> > We added a paragraph in section 5 to discuss the connection of reward fine-tuning and GLASS Flows. GLASS has 2 different important synergies with these models:
> > 1. **GLASS Flows can be applied to fine-tuned models:** One can also apply inference-time scaling with GLASS Flows to fine-tuned models. Many of these models learn a FM model of reward-tilted distribution. Hence, GLASS Flows is equally valid to be applied to reward fine-tuned models to improve them at inference-time.
> > 2. **GLASS could be used to accelerate fine-tuning:** Many reward fine-tuning algorithms - including the ones cited by the reviewer - require the use of DDPM/SDE sampling for exploration during training [1,2,3,4]. However, this is very inefficient - as we show in our GLASS Flows work. Therefore, one could potentially accelerate reward fine-tuning methods via GLASS Flows by replacing the slow SDE sampling with GLASS Flows. This is a completely new application of GLASS Flows that could be explored in the future. Partial replacement of SDEs with ODEs was already explored in [2] and we could extend this idea significantly with GLASS Flows.
> >
> > **Questions about Few-Step Sampling Results:**
> > - Poor performance for few-step sampling for SDEs: We double checked our code and our code was correct. The poor performance is mainly due to known poor performance of SDE sampling in the few-step regime, see e.g. [5].
> > - FID-vs-steps curve for standard ODE sampling: We did not compare against ODE sampling in section 6.1. as ODE sampling does not sample from the same posterior distribution and a comparison would be unfair. Section 6.2 focuses on sampling comparison against ODE.
> > - Other SDE solvers: We agree that there are many interesting and complex solvers for SDEs. However, there are also many higher-order ODE solvers. We use Euler sampling both for ODE and SDE sampling to be fair to both methods. We consider higher-order solvers as complementary to GLASS Flows (i.e. these solvers can be used to improve all existing methods).
> >
> > **Best-of-N Sampling with GLASS Flows (Table 1).**
> > We provided the requested values for Best-of-N sampling with GLASS Flows now in Table 1. The results are very similar to the results for Best-of-N sampling with ODE sampling, as expected by sampling results in 6.2. We hope this ablation allows us to provide an even stronger case for GLASS Flows as it shows that it is truly the stochastic evolution of GLASS Flows that can be exploited by FKS leading to state-of-the-art results.
> >
> >
> > [1] DanceGRPO: Unleashing GRPO on Visual Generation
> >
> > [2] MixGRPO: Unlocking Flow-based GRPO Efficiency with Mixed ODE-SDE
> >
> > [3] FlowGRPO: Flow-GRPO: Training Flow Matching Models via Online RL
> >
> > [4] Adjoint Matching: Fine-tuning Flow and Diffusion Generative Models with Memoryless Stochastic Optimal Control
> >
> > [5] Elucidating the Design Space of Diffusion-Based Generative Models

---

> ### Author Response · Authors · 2025-11-21
>
> > After fixing rho, how much variation do different initial noise samples produce in the final generated images?
>
> We briefly discuss the role of $\rho$ and how it determines the stochastic nature of the GLASS transition $p_{t'|t}(x_{t'}|x_{t})$. The GLASS transition can be written as:
> $$\begin{align*}
> p_{t'|t}(x_{t'}|x_{t})=\int p_{t'|t}(x_{t'}|x_{t},z)p_{1|t}(z|x_{t}) d z
> \end{align*}$$
> We note that $\rho$ determines the term $p_{t'|t}(x_{t'}|x_{t},z)$ but has no effect on the posterior $p_{1|t}(z|x_t)$. E.g. for $\rho=1$, the conditional transition $p_{t'|t}(x_{t'}|x_t,z)$ becomes deterministic. However, for $\rho=1$ the posterior is *not* deterministic but rather a proper posterior distribution. Therefore, even for $\rho=1$, the GLASS transitions are stochastic. This illustrates that $\rho$ has only an effect on one factor determining the stochastic nature of the transition. We found that the time difference $t'-t$ and the variance of the posterior $p_{1|t}$ are equally important factors determining the variance of the transition distribution $p_{t'|t}(x_{t'}|x_{t})$.
>
> > And how does the diversity change as K varies?
>
> $K$ are the total number of transitions in the paper. The number of transitions determine the time difference $t'-t$. This is a very important factor in the nature of the transitions. The larger $t'-t$ (this means the smaller $K$), the higher the diversity is. We hope this clarifies your question.
>
> Thank you for your interest in our work.

---

### Official Review · Reviewer_dnmw · 2025-11-14

**Soundness:** 4
**Presentation:** 3
**Contribution:** 3
**Rating:** 6
**Confidence:** 5

**Summary:**

Sorry for my late review!

This paper addresses the inefficiency of inference-time reward adaptation for flow matching and diffusion models, which currently require sampling stochastic Markov transitions \(p_{t'|t}(x_{t'} \mid x_t)\) via SDEs even though ODE sampling is far more efficient but deterministic, thereby inducing a trade-off between stochastic exploration and runtime. The authors propose Gaussian Latent Sufficient Statistic (GLASS) Flows, which construct an “inner” flow matching model that realizes a rich family of Gaussian Markov transitions (including DDPM transitions as a special case) as ODE trajectories in an auxiliary time \(s\), enabling ODE-based sampling of stochastic transitions directly from a pre-trained model. Technically, the method treats \((x_t, x_{t'})\) as two noisy measurements of a latent variable \(z\), compresses them into a sufficient statistic \(S(x_t, x_{t'})\), and shows that the standard marginal denoiser \(D_t\) of the base model can be reused—with an appropriate input and time reparameterization—to define a GLASS denoiser and the corresponding conditional velocity field \(u_s(x_s \mid x_t; t)\) without any retraining. The paper proves that integrating this inner ODE from \(s=0\) to \(s=1\) samples from the desired GLASS transition in the continuous-time limit, analyzes discretization error, and argues that GLASS Flows inherit the computational efficiency and stability of ODE samplers while providing controlled stochastic evolution comparable to SDEs. Empirically, the authors first show on base text-to-image generation with SiT and FLUX that GLASS-based transitions match ODE sampling in FID and alignment metrics and substantially outperform DDPM-style SDE sampling at matched compute, thus eliminating the efficiency–stochasticity trade-off at the base model level. They then plug GLASS Flows into Sequential Monte Carlo Feynman–Kac steering for inference-time reward adaptation on FLUX, across four reward models (CLIP, PickScore, HPSv2, ImageReward) and benchmarks such as GenEval and PartiPrompts, demonstrating that FKS with standard SDE transitions fails to beat a simple Best-of-\(N\) ODE baseline whereas FKS–GLASS consistently improves both the target reward and GenEval scores without extra cost. Finally, combining GLASS-based FKS with gradient-based reward guidance further boosts reward scores and alignment metrics, showing that GLASS Flows serve as a simple, theoretically grounded, and practically effective drop-in replacement for SDE sampling that enables more powerful and efficient inference-time reward alignment of large text-to-image flow and diffusion models.

**Strengths:**

- The paper introduces a method for inference-time reward adaptation by reframing stochastic Markov transitions as an inner flow matching problem and leveraging a Gaussian latent sufficient statistic to re-use existing denoisers, yielding a unified ODE-based mechanism that subsumes DDPM-style transitions instead of proposing yet another bespoke reward adaptation algorithm.
- The technical development is careful and well-grounded, deriving GLASS transitions from a joint Gaussian model, justifying the sufficient-statistic construction, proving that the induced velocity field satisfies the continuity equation, and supporting the method with extensive experiments on both base generation and multiple reward-adaptation setups, including compute-matched comparisons against strong ODE/SDE baselines.
- The exposition is generally clear and well structured: the ODE–SDE trade-off is crisply articulated, the roles of marginal vs. conditional flows are distinguished, the GLASS denoiser is given an intuitive “two noisy measurements” interpretation, and the paper repeatedly connects theory to practical design choices (e.g., how to plug GLASS into FKS and reward guidance).

**Weaknesses:**

The core technical step in GLASS Flows—the use of a linear sufficient statistic over multiple Gaussian observations to recover the original denoiser at a reparameterized time—appears mathematically very close to the training-equivalence result in TADA (arXiv:2506.21757), in particular Proposition 3.1. In GLASS, Proposition 2 shows that the GLASS denoiser (D_{\mu,\Sigma}(x_t, x_{t'})) can be written as the original denoiser evaluated at a suitably reweighted statistic and an effective time determined by (\mu^\top \Sigma^{-1}\mu); TADA’s Proposition 3.1 establishes an essentially identical form for an augmented Gaussian state, again via a reweighted linear combination and an effective SNR. Since the underlying Gaussian-conjugacy/sufficient-statistic argument is the same, it would be helpful if the authors could explicitly cite TADA and briefly discuss how their Proposition 2 relates to Proposition 3.1 there, while emphasizing that GLASS applies this mechanism in a different context (Markov transitions and reward alignment).


1) In Equation (1), the transition is defined as $p_{t'|t}(x_{t'}|x_t)=\mathbb{P}[X_{t'}=x_t|X_t=x_t]$. The left-hand variable in the probability should be $x_{t'}$ rather than $x_t$.

2) In Algorithm 1 (left panel), the function `D(x,t)` returns:
  - `(1 / (dot_alpha_t * sigma_t - alpha_t * dot_sigma_t)) * (sigma_t * u - dot_sigma_t)`
- The factor of $x$ multiplying $\dot{\sigma}_t$ seems to be missing. The standard denoiser identity is:
  - $D_t(x)=\frac{1}{\dot{\alpha}_t\sigma_t-\alpha_t\dot{\sigma}_t}\left(\sigma_t u_t(x)-\dot{\sigma}_t x\right)$.

3) In the GLASS denoiser, $S(\mathbf{x})=\frac{\mu^\top\Sigma^{-1}}{\mu^\top\Sigma^{-1}\mu}[x_t,\bar x_s]^\top$. With the natural (CondOT) choice $\bar\alpha_0=0$, we have $\mu=(\alpha_t,\,\bar\alpha_s+\bar\gamma\alpha_t)^\top$. When $t=0$ and $s=0$, both $\alpha_t=0$ and $\bar\alpha_s=0$, resulting in $\mu=\mathbf{0}$ and potential division by zero in both $S(\x)$ and $(\mu^\top\Sigma^{-1}\mu)^{-1}$. This edge case arises in the first Euler call at $s=0$ in Algorithm 1 (right panel), and should be handled explicitly in the implementation. The correct limiting behavior at $s=0$ is that $\bar X_0$ carries no additional information about $Z$ beyond $X_t$ (since $\bar X_0=\bar\gamma X_t+\bar\sigma_0\epsilon$ does not depend on $Z$ given $X_t$). Thus:
  - $\lim_{s\to 0} D_{\mu(s),\Sigma(s)}(x_t,\bar x_s) = D_t(x_t)$.

4) Since $p_{t'|t}$ is constructed from a joint $p_{t,t'}$ with correct marginals $p_t$ and $p_{t'}$, for any fixed pair $(t,t')$:
  - $\int p_{t'|t}(x'|x) p_t(x)\,dx = p_{t'}(x')$.
- Therefore, under exact integration of the inner ODE, composing GLASS transitions across a grid $0=t_0<\cdots<t_K=1$ should produce $X_{t_k}\sim p_{t_k}$ at every step (and hence $X_1\sim p_1$), for any $\rho\in[-1,1]$. This is an important property: GLASS sampling preserves correct marginals even when $\rho$ differs from the DDPM value (though the multi-time joint may differ from that of a particular SDE/ODE). Since this property is central to GLASS as a sampling scheme, the paper would benefit from making it explicit with a brief proof in the main text or appendix.

5) In Appendix A.1, the GLASS guidance is formulated as:
  - $u_s^r(\bar x_s|x_t,t)=u_s(\bar x_s|x_t,t)+\beta_{t^\star}\frac{\sigma_{t^\star}^2}{\alpha_{t^\star}}w_2(s)\nabla_y r(D_{t^\star}(y))\big|_{y=\alpha_{t^\star}S(\x)}$.
- This follows from replacing $D_{\mu,\Sigma}$ with $D_{\mu,\Sigma}^r=D_{t^\star}^r(\alpha_{t^\star}S(\x))$ in the $w_2(s)D$ term, treating guidance as a "denoiser-level" correction. However, readers familiar with $u_t^r=u_t+c_t\nabla V_t(x)$ in the standard ODE domain may find this formulation unclear. A brief derivation showing that GLASS guidance is the "denoiser-based" guidance analog of $D_t^r=D_t + \frac{\sigma_t^2}{\alpha_t}\nabla V_t$ transplanted into the $w_2(s)D$ structure would clarify the construction.


6) For $\rho=\frac{\alpha_t\sigma_{t'}}{\sigma_t\alpha_{t'}}$, the paper claims that $p_{t'|t}^{\text{DDPM}}=p_{t'|t}^{\text{GLASS}}$. The proof in Appendix A.2 is sound, computing the conditional of the forward noising process (OU skeleton) and matching covariances. One minor note: the paper should explicitly state that $|\rho|\le 1$ holds in this case, since $\alpha_t/\alpha_{t'}<1$ and $\sigma_{t'}/\sigma_t<1$ for $t<t'$.





7) Each GLASS inner step requires one neural network call to $u$ (then $D$ via the reparameterization) and a few $O(1)$ operations for the 2×2 inverse, $t^\star$ lookup/inversion of $g$, and forming $S(\x)$. This is comparable to the SDE per-step cost (one NN call), supporting the claim that GLASS can be swapped into SMC at essentially zero additional cost.

8) Computing $D_t$ from $u_t$ requires $\dot\alpha_t,\dot\sigma_t$. Many deployed models expose continuous-time schedulers, but some libraries wrap discrete-time models and approximate derivatives. The paper should mention the requirement that $\alpha_t,\sigma_t$ be differentiable and clarify how discrete-time models are handled in practice.

9) GLASS transitions achieve stochasticity by sampling the inner initial condition $\bar X_0\sim \mathcal{N}(\bar\gamma x_t,\bar\sigma_0^2 I)$, while the subsequent evolution follows a deterministic ODE. Clarifying this distinction would help readers who might associate "stochastic transitions" exclusively with SDEs.

[1] Chen, T., Zheng, H., Berthelot, D., Gu, J., Susskind, J., & Zhai, S. (2025). TADA: Improved Diffusion Sampling with Training-free Augmented Dynamics. arXiv preprint arXiv:2506.21757.

**Questions:**

The weight function $w_1(s)=\partial_s\bar\sigma_s/\bar\sigma_s$ can become numerically unstable if $\bar\sigma_s$ approaches zero. In your setup, $\bar\sigma_1^2=\sigma_{t'}^2(1-\rho^2)$, which remains positive unless $\rho=\pm 1$. Since the method allows $\rho\in[-1,1]$, the paper should warn users that $\rho=\pm 1$ leads to degenerate cases where $\bar\sigma_1=0$ and $w_1$ becomes large near $s=1$, which could be numerically problematic with Euler integration. In practice, should users avoid $\rho$ values too close to $\pm 1$?

---

> ### Author Response · Authors · 2025-11-21
>
> We sincerely thank the reviewer for insightful comments including the constructive feedback allowing us to improve our work.
>
> **Relation to TADA [1].**
>
> >It would be helpful if the authors could explicitly cite TADA and briefly discuss how their Proposition 2 relates to Proposition 3.1. there.
>
> We thank the reviewer for making us aware of this closely related work. While both works introduce different algorithms, we agree that the same mathematical tools of Gaussian conjugacy/sufficient statistics are indeed applied to achieve training-free improvements. We made several changes to highlight the connection in the new version of our work:
> - Related work section: We have added a paragraph in the related work section giving a high-level description of the mathematical connection between both works (see section 5).
> - Extended related work section: We have added a more detailed discussion of how both works use the principle of sufficient statistics (see appendix C).
>
> Finally, we would like to mention that the official ICLR policy counts concurrent work as up to 4 months before submission and TADA was released roughly 4 months before submission. As not citing TADA was mentioned as the main weakness of our submission, we would appreciate it if this could be taken into account.
>
> **Numerical stability.** Thank you for addressing numerical stability. We added a section B.3. explaining the exact cases when numerical instability might occur and how this can be easily resolved.
> - **s=0 case:**
> >This edge case arises in the first Euler call at in Algorithm 1 (right panel), and should be handled explicitly in the implementation.
>
> We derive the correct form for the denoiser for $s=0$ in this case in appendix B.3. Further, as suggested by the reviewer, we have accounted for the edge case of s=0 in algorithm 1 now.
>
> - **$\rho=\pm 1$ case:**
> > In practice, should users avoid values rho close to rho=+-1?```
>
> Users do not have to avoid values of rho=+-1. To showcase this, we have also done additional experiments for rho=1 to show that this value also leads to high-quality samples with the FLUX model (see section D.3). We discuss this case in detail in appendix B.3.
>
> **Other comments:** Thank you for your constructive and extensive comments allowing us to improve the clarity of our work. We explain how we addressed your comments in the new version:
> 1. Transition in equation (1): Thank you for spotting this typo. We corrected it.
> 2. Denoiser missing $x$ in algorithm: Thank you for spotting this typo. We corrected it.
> 3. Case for $s=0$: see “concerns on numerical stability” above and new section in appendix B.3.
> 4. Preservations of marginals: We have added a paragraph “Sampling with GLASS Flows” in section 4.2 that makes this point clearer.
> 5. Gradient guidance: We rewrote section B.1 explaining the gradient guidance as a translation of the known construction. We hope that this makes the derivation clearer.
> 6. Valid correlation parameter: We have added a comment after proposition 1.
> 7. Stressing negligible overhead in algorithm 1: We explained that everything in algorithm 1 is negligible except the neural network calls in section 4.2.
> 8. Discrete-time diffusion models: We have explicitly added the assumption that schedulers are differentiable. Further, we wrote an additional section in appendix B.4 to explain how to handle discrete-time diffusion parameterizations. We stress that discrete-time diffusion models can be used in the same way as long as we restrict ourselves to a constrained set of inner time points $s$ in the GLASS Flow simulation (see details in appendix B.4).
> 9. Source of stochasticity: We stressed this core principle in the beginning of section 4 now (lines 183-185).
>
> **Questions.** see paragraph “numerical stability” above.
>
> Thank you again for your helpful feedback and response. We welcome any additional discussions.
>
> [1] Chen, T., Zheng, H., Berthelot, D., Gu, J., Susskind, J., & Zhai, S. (2025). TADA: Improved Diffusion Sampling with Training-free Augmented Dynamics. arXiv preprint arXiv:2506.21757.

---

### Author Response · Authors · 2025-11-21

We would like to thank all reviewers for their constructive and positive feedback. We are pleased to see that our work is viewed as “a fundamental improvement” in the development of diffusion models and appreciated for its “elegance,” “broad applicability,” and “drop-in practicality.”

**Summary of updates.** For convenience, we summarize the most important updates and additional experiments incorporated into the revised manuscript:
- Ablation on the correlation parameter $\rho$ for GLASS transitions, clarifying the choice of $\rho=0.4$ (Appendix D.3).
- Best-of-N sampling experiments demonstrating that GLASS’s stochasticity is key to its SOTA performance (Table 1).
- Diversity metrics across all methods, further illustrating the sampling behavior (Appendix D.2).
- Expanded related work, including detailed discussion of closely-related works and reward-fine-tuning methods (Section 5, Appendix C).
- Numerical stability analysis, including practical guidance and updates to Algorithm 1 (Appendix B.3).
- Additional diffusion parameterizations, showing how GLASS applies to discrete-time or alternative non-FM formulations (Appendix B.4).

All reviewer comments have been addressed in detail in the updated manuscript on OpenReview as well as in the individual point-by-point responses. We thank the reviewers for the time and feedback.

---

### Meta-Review · Area_Chair_JdoC · 2025-12-21

**Summary:**

This paper introduces GLASS Flows, a new sampling paradigm that simulates a “flow matching model within a flow matching model” to sample Markov transitions. The approach aims at sampling diffusion models with ODE solver's efficiency as well as SDE solver's diversity. In the initial review, reviewers rate this paper with score 6, 8, 6 and 8. The concerns mainly lie in the discussion with related work (Reviewer dnmw and 56nT), parameter analysis and numerial stability (Reviewer 1Ktu and 56nT), the difference from RL (Reviewer JMLt and 56nT), and the writing (Reviewer dnmw).

During rebuttal, authors provides response to each question and most concerns are addressed. Overall, AC confirms that the experiment is extensive, and the results are impressive. This paper receives all positive rating in the initial review with average score of 7, and thus AC recommends this paper to **Accept (Oral)**.

**Reviewer Concerns:**

The initial concerns mainly lie in the discussion with related work (Reviewer dnmw and 56nT), parameter analysis and numerial stability (Reviewer 1Ktu and 56nT), the difference from RL (Reviewer JMLt and 56nT), and the writing (Reviewer dnmw).

During rebuttal, authors provide explanation to the related works, conduct additional analysis on the parameter $\rho$, and add a new section on the numerical stability analysis. The authors also explain the difference from RL algorithms and how this work applies to RL post-trained models. Despite Reviewer dnmw, JMLt and 56nT unable to reply, AC confirms that the concerns are mostly addressed.

**Reviewer Scores:**

**Reviewer dnmw**: keep 6.
Reviewer dnmw does not participates in the discussion, but his concerns and questions are all responded by the authors. AC thinks his concerns are mostly addressed.

**Reviewer JMLt**: keep 8.
Reviewer JMLt does not participates in the discussion, but his concerns and questions are all responded by the authors. AC thinks his concerns are mostly addressed.

**Reviewer 56nT**: keep 6.
Reviewer 56nT does not participates in the discussion, but his concerns and questions are all responded by the authors. AC thinks his concerns are mostly addressed.

**Reviewer 1Ktu**: keep 8. Reviewer 1Ktu participates in the discussed and decided to keep the rating.

---

### Decision · Program_Chairs · 2026-01-26

Accept (Oral)